# PATHGEN-1.6M: 1.6 MILLION PATHOLOGY IMAGE-TEXT PAIRS GENERATION THROUGH MULTI-AGENT COLLABORATION

**Yuxuan Sun**[1,2]  **Yunlong Zhang**[1,2]  **Yixuan Si**[2]  **Chenglu Zhu**[2]  **Kai Zhang**[3]
**Zhongyi Shui**[1,2]  **Jingxiong Li**[1,2]  **Xuan Gong**[4]  **Xinheng Lyu**[2]  **Tao Lin**[2*]  **Lin Yang**[2,5*]

[1] Zhejiang University   [2] Westlake University
[3] Ohio State University   [4] Harvard University
[5] Center for Interdisciplinary Research and Innovation, Muyuan

## ABSTRACT

Vision Language Models (VLMs) like CLIP have attracted substantial attention in pathology, serving as backbones for applications such as zero-shot image classification and Whole Slide Image (WSI) analysis. Additionally, they can function as vision encoders when combined with large language models (LLMs) to support broader capabilities. Current efforts to train pathology VLMs rely on pathology image-text pairs from platforms like PubMed, YouTube, and Twitter, which provide limited, unscalable data with generally suboptimal image quality. In this work, we leverage large-scale WSI datasets like TCGA to extract numerous high-quality image patches. We then train a large multimodal model (LMM) to generate captions for extracted images, creating PathGen-1.6M, a dataset containing 1.6 million high-quality image-caption pairs. Our approach involves multiple agent models collaborating to extract representative WSI patches, generating and refining captions to obtain high-quality image-text pairs. Extensive experiments show that integrating these generated pairs with existing datasets to train a pathology-specific CLIP model, PathGen-CLIP, significantly enhances its ability to analyze pathological images, with substantial improvements across nine pathology-related zero-shot image classification tasks and three whole-slide image tasks. Furthermore, we construct 200K instruction-tuning data based on PathGen-1.6M and integrate PathGen-CLIP with the Vicuna LLM to create more powerful multimodal models through instruction tuning. Overall, we provide a scalable pathway for high-quality data generation in pathology, paving the way for next-generation general pathology models. Our dataset, code, and model are open-access at PathGen-1.6M.

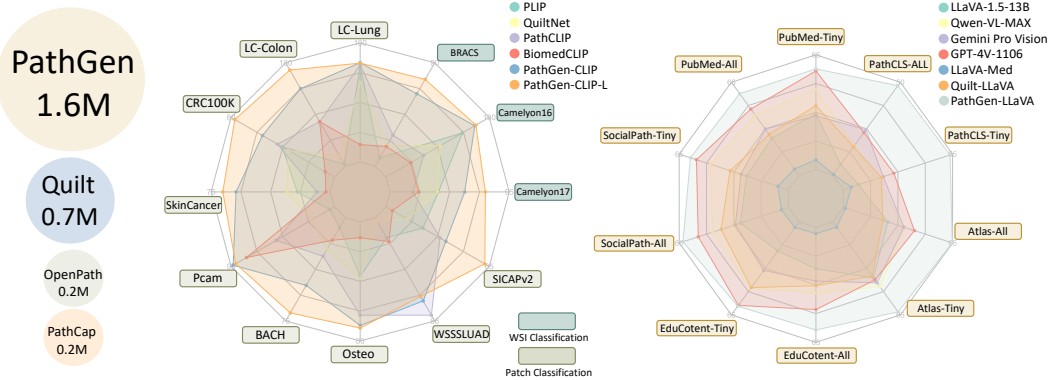

Figure 1: Illustration of the scale of the PathGen dataset (left), the performance of the proposed PathGen-CLIP (middle), and the PathGen-LLaVA (right), both derived from training on PathGen.

---

*Corresponding authors.

# 1 INTRODUCTION

Pathology plays a crucial role in modern medicine as it is the gold standard for disease diagnosis and the selection of treatment methods (Kumar et al., 2014). With the rapid growth of artificial intelligence, there is an increasing interest in developing robust general-purpose models to assist physicians, particularly in pathology. Pathology-specific CLIP models have demonstrated exceptional performance in zero-shot image classification (Radford et al., 2021; Jia et al., 2021), multimodal understanding (Liu et al., 2024; 2023; Bai et al., 2023; Li et al., 2022; 2023b; Dai et al., 2024), robustness to various perturbations (Radford et al., 2021; Shu et al., 2023; Zhou et al., 2022), and scalability across diverse tasks (Rombach et al., 2022; Lin et al., 2023b; Esmaeilpour et al., 2022).

However, training such models typically requires vast amounts of data. For instance, general CLIP models are trained using massive amount of data from sources like WIT (Radford et al., 2021) and LAION (Schuhmann et al., 2022a), at scales of millions or even billions. In pathology, researchers are similarly focused on amassing large collections of pathology image-text pairs from various sources, including academic articles from PubMed (Lin et al., 2023a; Ikezogwo et al., 2024; Sun et al., 2024b), social media (Huang et al., 2023; Ikezogwo et al., 2024), and books (Gamper & Rajpoot, 2021).

Despite these efforts, the largest datasets do not exceed one million samples, which is significantly smaller compared to the scale of natural image datasets. This underscores the challenges in the availability and scalability of pathology-related datasets. Key limitations include: (1) The available pool of image-text pairs from the internet and books are quickly exhausted, hindering scalability. (2) Many collected images suffer from significant quality loss during the acquisition process. For example, images from PubMed articles often undergo compression, while educational content from platforms like YouTube typically consists of screenshots at 1080p resolution, further degraded by video compression, making them incomparable to the high-resolution images used in practical scenarios. (3) Unpaired image-text pairs frequently appear on social media, where users may post pathological images with comments focusing on non-pathological features like aesthetic appeal.

Fortunately, The Cancer Genome Atlas (TCGA) is a comprehensive, publicly funded project that provides clinical data across various cancer types. This dataset includes numerous WSIs, high-resolution scans from patient tissue samples. These WSIs inherently contain an immense amount of detailed information—such as cellular structures, tissue organization, and morphological patterns—crucial for cancer diagnosis and research. However, these datasets typically only provide labels at the slide level, leaving patches within WSIs without detailed textual annotations. This lack of specific annotations hinders models from learning rich semantic information from such high-quality image data.

In this study, we aim to harness high-quality images from WSIs to construct a large-scale image-text dataset. We develop a cascaded approach involving multiple agent models that collaborate to extract the most representative patches from WSIs and generate captions describing the visual details within each patch. This process enables us to compile a dataset of 1.6 million image-caption pairs, designed to train and enhance pathology-specific multimodal models. **Our main contributions are as follows: (1)** We propose PathGen-1.6M, which is currently the largest and highest-quality pathology image-text dataset, as depicted in Figure 1. **(2)** The construction approach of PathGen-1.6M offers a scalable solution to expand the currently limited pool of pathology image-text pairs, addressing a critical need in the field. **(3)** Experiments confirm that PathGen-1.6M can significantly boost the performance of existing multimodal models like CLIP and LLaVA in the pathology domain, while scaling up PathGen presents a promising potential as a future pretraining strategy for pathology models.

# 2 RELATED WORK

**Existing Vision-language Datasets.** Training vision-language models like CLIP (Radford et al., 2021) requires large and high-quality image-text pairs to capture the richness of visual and semantic information. In the general domain, notable datasets have been constructed such as LAION-5B (Schuhmann et al., 2022a)and WIT-400M (Radford et al., 2021). In pathology, ARCH (Gamper & Rajpoot, 2021) and PathCap (Sun et al., 2024b) datasets are collected from PubMed and medical textbooks, containing 11,816 and 207,000 pathology image-caption pairs, respectively. The OpenPath (Huang et al., 2023) dataset includes 208,414 pairs from Twitter posts, while the Quilt-1M (Ikezogwo et al., 2024) dataset gather 768,826 histopathology image-text pairs from video frames and corresponding subtitles on YouTube. These datasets are primarily gathered from social media platforms and text-

books. Although fine-tuning CLIP on these specialized pathology datasets significantly enhances its adaptation to pathology tasks, challenges such as low image quality, image-text misalignment, and poor data scalability continue to hinder the further development of pathology-specific CLIP models.

**Pathology-specific CLIP Models.** CLIP (Radford et al., 2021) is a powerful model that learns visual concepts through natural language supervision. In general domains, it demonstrates remarkable capabilities in zero-shot classification, image retrieval, and multimodal understanding by training vast image-caption datasets. This approach leverages language's semantic richness to enhance visual recognition, making it adaptable across various applications without task-specific training. In pathology domain, CLIP's potential is being increasingly recognized as a solution to major challenges, such as the scarcity of labeled data and the requirement for domain-specific expertise. By leveraging natural language descriptions, CLIP facilitates the identification and classification of intricate features that traditionally pose significant challenges for manual annotation. Recently, there has been a notable increase in pathology CLIP variants, including PubMedCLIP (Eslami et al., 2023), BiomedCLIP (Zhang et al., 2023a), PMC-CLIP (Lin et al., 2023a), Quilt-Net (Ikezogwo et al., 2024), PathCLIP (Sun et al., 2024b), PLIP (Huang et al., 2023), and CONCH (Lu et al., 2024).

**Large Multimodal Model (LMM).** The integration of large language models (LLMs) like GPT-3 (Brown et al., 2020), T5 (Raffel et al., 2020), and GPT-4 (OpenAI, 2023a) with vision capabilities has spurred the development of sophisticated multimodal models (LMMs). These LMMs, such as Flamingo (Alayrac et al., 2022), BLIP-2 (Li et al., 2023b), and Fuyu (Bavishi et al., 2023), excel in multimodal understanding by utilizing pretraining techniques. Additionally, instruction-tuning, derived from NLP, has been adapted for LMMs, enabling them to generate more controllable and task-specific outputs through datasets like those used in GPT-4V (OpenAI, 2023b), Gemini Pro Vision (Team et al., 2023), Qwen-VL (Bai et al., 2023), and InstructBLIP (Dai et al., 2023). The application of LMMs in pathology is particularly promising. Models such as PathAsst (Sun et al., 2024b), LLaVA-Med (Li et al., 2023a), Quilt-LLaVA (Seyfioglu et al., 2023) have been developed using curated pathology-specific instruction-tuning datasets sourced from resources like PubMed and educational YouTube videos. These advancements facilitate effective analysis and the generation of descriptive texts for pathological images. Consequently, we leverage this capability to generate corresponding descriptions for image patches within WSIs. By creating high-quality image-text pairs, we aim to enhance the foundational vision-language models in pathology.

**Multi-Agent Collaboration.** With the advancement of large models (LMs) and the development of specialized models for various tasks, recent research has explored the use of multi-agent collaboration. This approach allows these models to work together, achieving tasks that are beyond the capabilities of any single model alone. For instance, leveraging LLMs for role-playing can be used to accomplish tasks such as software development (Hong et al., 2023; Qian et al., 2023), societal simulation (Park et al., 2023; 2022), policy simulation (Xiao et al., 2023; Hua et al., 2023), game simulation (Xu et al., 2023; Wang et al., 2023b) and video generation (Yuan et al., 2024).

## 3 PATHGEN DATASET CONSTRUCTION

The entire data construction pipeline is illustrated in Figure 2. We employ multiple agents working collaboratively to generate high-quality pathology image-text pairs. This process involves extracting representative WSI image patches through CLIP-based image retrieval and clustering. These patches are then described by a trained pathology LMM agent, followed by another LMM agent and an LLM agent that revises and summarizes the descriptions. In this section, we detail the construction of the agent model and explain their collaborative workflow in the data generation process.

### 3.1 AGENT MODEL PREPARATION

**PathGen-CLIP-L$_{init}$:** General models like OpenAI's CLIP underperform in the pathology domain, necessitating a specialized model for tasks such as cross-modal retrieval we employ in section 3.2. For this purpose, we combine portions of existing datasets, including PathCap (200K), as well as cleaned versions of Quilt-1M (400K) and OpenPath (100K), where misaligned image-caption pairs are removed from the latter two datasets, resulting in a total of 700K samples. We refer to this dataset as PathGen$_{init}$. Utilizing the OpenCLIP framework (Ilharco et al., 2021), we train a CLIP-L version of the pathology-specific CLIP model with a 336 input image size, designated as PathGen-CLIP-L$_{init}$.

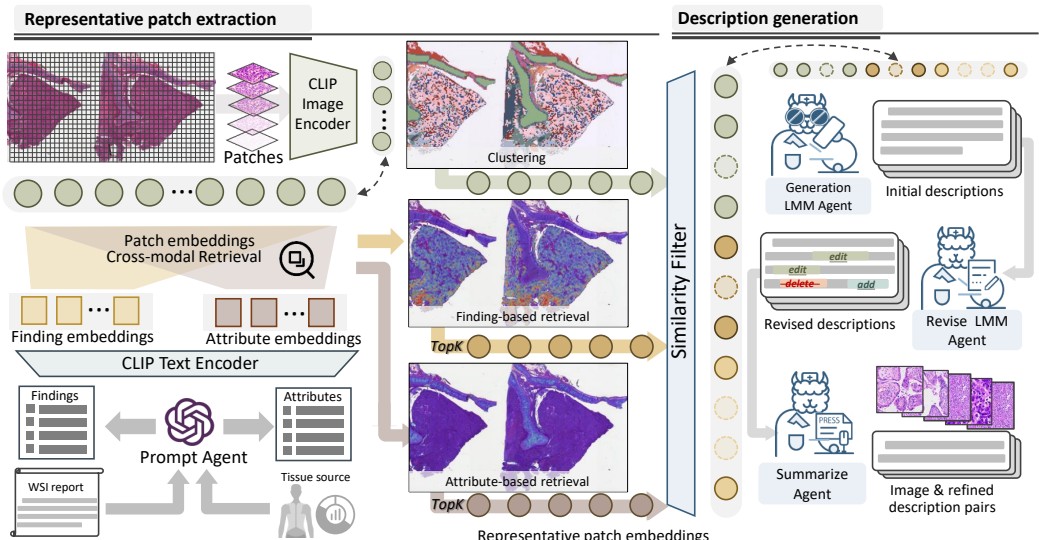

Figure 2: Illustration of the multi-agent collaboration pipeline for generating pathology image-text pairs. This process comprises two main components: (1) Representative Patches Extraction, which utilizes prompt-based cross-modal retrieval and clustering; and (2) Description Generation, where multiple LMM and LLM agents are employed to generate, revise, and summarize descriptions.

**Description LMM Agent:** To generate high-quality pathology image-text pairs, we require a pathology-specific LMM capable of producing accurate and detailed image captions. Existing image-caption pairs are often too simplistic to comprehensively describe these details, hindering the training of more effective description generation models. Inspired by the PathMMU benchmark (Sun et al., 2024a), we sample 30,000 image-caption pairs, with 10,000 pairs each from PathCap, OpenPath, and Quilt-1M. We provide each image along with its corresponding caption to GPT-4V, allowing the model to enhance and refine the original captions by incorporating details observed in the images. This approach enables us to generate 30,000 detailed image descriptions. We build upon LLaVA-v1.5-13B (Liu et al., 2024) by replacing its OpenAI-CLIP vision encoder with our PathGen-CLIP-L$_{init}$. We then trained this modified architecture on our curated pathology image-descriptions pairs to create PathGen-LLaVA$_{desp}$, our pathology-specific description LMM agent. We compare the pathology image description generation capabilities of PathGen-LLaVA$_{desp}$ with those of LLaVA-Med (Li et al., 2023a) and Quilt-LLaVA (Seyfioglu et al., 2023) in Appendix B.1.

**Revise LMM Agent**: The Revise Agent is a pathology LMM built on the LLaVA-v1.5-13B framework (Liu et al., 2024), designed with error-correction capabilities. While existing pathology LMMs are trained only for limited tasks, such as multiple-choice questions, dialogue, and description, they lack the capabilities to perform self-correction. To achieve this, we leverage descriptions generated by the description LMM Agent and prompt GPT-4 to systematically introduce controlled inaccuracies through three defined operations: add, delete, or edit. By reversing these operations—adding becomes deleting, deleting becomes adding, and edits are inverted—enables us to create a multimodal caption editing dataset of (image, caption, editing operation) triplets. These derived operations, pre- and post-modification descriptions, and corresponding images are then employed to train the Revise LMM Agent, thus equipping it with robust multimodal error-correction capabilities.

**Summarize Agent:** Due to the CLIP model's limitation of accepting only 77 tokens as input, the data generated by PathGen-LLaVA$_{desp}$ often exceeds this length. To address this, we prompt GPT-4 to generate instruction-tuning data for summarizing these descriptions. We then fine-tune Llama-2 as a summary agent to produce concise summaries for each generated description of WSI patches.

## 3.2 DETAILS OF DATA CONSTRUCTION PIPELINES

**Source Data:** We source approximately 7,500 WSIs with paired reports from TCGA. Since these reports often contain information unrelated to WSIs, such as gross findings and measurements. Inspired by HistGen (Guo et al., 2024), we prompt GPT-4 to extract only observable morphological

Table 1: Human evaluation of image captions from PathGen-1.6M and OpenPath. The accuracy represents the proportion of correct findings in each caption on average.

| Pathologist ID | PathGen-1.6M | | | | | | OpenPath | | | | | |
|---|---|---|---|---|---|---|---|---|---|---|---|---|
| | Number of Captions | Total Findings | Findings per Caption | Correct Findings | Incorrect Findings | Accuracy | Number of Captions | Total Findings | Findings per Caption | Correct Findings | Incorrect Findings | Accuracy |
| Pathologist A | 200 | 1,059 | 5.30 | 956 | 103 | **90.3%** | 200 | 412 | 2.06 | 319 | 93 | 77.4% |
| Pathologist B | 200 | 1,059 | 5.30 | 937 | 122 | **88.5%** | 200 | 412 | 2.06 | 312 | 100 | 75.7% |

and diagnostic features in reports. As some reports contain substantial information and may exceed the 77-token input limit for models like CLIP, we design GPT-4 prompts (Figure 18) to split longer reports into 2-3 concise sentences, while preserving critical information. With these WSIs and cleaned reports, we develop a five-step meticulous pipeline, as shown in Figure 2, spanning from representative patch extraction to description generation, to produce high-quality image-text pairs.

**Step 1: Representative Patch Extraction:** This step identifies representative patches from WSIs using prompt-based retrieval and k-means clustering. Prompt-based retrieval focuses on diagnostically relevant patches using WSI reports and predefined prompts, while k-means clustering ensures feature diversity by capturing distinct morphological patterns. Prompt-based retrieval uses PathGen-CLIP$_{init}$-L to identify relevant patches through two prompt types: (1) previously cleaned WSI reports, formatted as 1-3 concise paragraphs, as prompts. (2) GPT-4 generated prompts based on the WSI's tissue origin (e.g., lung, colon), describing potential attributes like enlarged nuclei or lymphocyte infiltration. For both report-based and attribute-based prompts, we identify the top 64 relevant patches in each WSI, resulting in 128 patches in total. Detailed GPT-4 prompts are provided in Appendix C.2. *(2) K-means clustering:* Since the prompt-based retrieval primarily focuses on patches with a higher degree of pathological changes, it may overlook patches with other morphological variations. To address this, we use PathGen-CLIP-L$_{init}$ to extract features from WSI patches and apply k-means clustering on these features. Each cluster represents patches with distinct morphological characteristics. The number of clusters is determined by the square root of the total number of patches, as larger WSI typically contains more distinct morphological features. We sample 256 patches from each WSI using clustering, ensuring uniform sampling within each cluster across the WSI to guarantee a more diverse selection of samples. Finally, we combine the extracted patches from prompt-based retrieval and clustering, resulting in a total of 384 representative patches sampled from each WSI.

**Step2: Similar Patch Filtering:** Since the representative samples may still contain highly similar patches, which may impede the subsequent training in CLIP contrastive learning. We utilize PathGen-CLIP$_{init}$ to compute patch similarities and exclude redundant patches within each WSI. Using a similarity threshold of 0.88, we employ a probabilistic filtering approach where patches exceeding this threshold are removed with a probability proportional to their similarity score.

**Step3: Description Generation:** We utilize the trained PathGen-LLaVA$_{desp}$ model with the prompt: "This is a histopathology image from source, describe this image in detail," where source indicates the origin of the WSI (e.g., lung, colon). This approach generates detailed descriptions for all representative patches extracted from the WSIs, thereby creating initial image-description pairs.

**Step4: Description Revision:** To mitigate potential errors and hallucinations in LMM-generated descriptions, we implement a trained revision agent that reviews image-description pairs and refines the description through additions, modifications, or removals, while preserving descriptions that are already accurate. This step ensures the overall quality of descriptions.

**Step5: Description Summarization:** Since the descriptions generated by PathGen-LLaVA$_{desp}$ are usually lengthy and often exceed the CLIP 77-token limit, we use a summarization agent to extract the key information from these descriptions, ensuring that no essential details are lost.

Through these steps, we generate a total of 1.6 million high-quality image-text pairs from 7,500 WSIs, sourced from 27 different tissue types. These diverse pathology image-text pairs are utilized for the contrastive learning pre-training of the CLIP model. This effort aims to develop a more robust pathology-specific CLIP model, which enhances support for downstream pathology tasks.

### 3.3 EXPERT EVALUATION OF CAPTION QUALITY IN PATHGEN-1.6M

To validate the accuracy of our model's captions, we conduct evaluations by randomly selecting 200 generated captions and breaking them down into individual pathological findings. Two expert

Table 2: Comparison of different CLIP models on zero-shot pathology image classification datasets with accuracy (%). The top performance is highlighted in **bold**, with the second-best underlined.

| Model | LC-Lung | LC-Colon | CRC100K | SkinCancer | Pcam | BACH | Osteo | WSSSLUAD | SICAPv2 | Average |
|-------|---------|----------|---------|------------|------|------|-------|----------|---------|---------|
| OpenAI-CLIP | 33.1 | 75.7 | 26.2 | 9.6 | 53.9 | 21.7 | 46.9 | 64.6 | 32.8 | 40.6 |
| OpenAI-CLIP-L | 70.4 | 81.1 | 40.3 | 19.4 | 55.5 | 34.3 | 53.9 | 81.2 | 25.4 | 51.3 |
| PLIP | 87.9 | 90.2 | 52.8 | 42.5 | 51.8 | 34.3 | 52.9 | 73.1 | 42.5 | 58.6 |
| PubmedCLIP | 33.3 | 80.5 | 31.5 | 11.3 | 65.4 | 34.8 | 30.0 | 65.4 | 7.0 | 39.8 |
| PMC-CLIP | 33.3 | 51.9 | 8.7 | 11.4 | 53.8 | 21.3 | 29.2 | 65.2 | 31.5 | 34.0 |
| QuiltNet | 80.0 | 91.0 | 49.5 | 46.4 | 58.7 | 43.8 | 53.8 | 70.5 | 37.3 | 58.9 |
| PathCLIP | 88.9 | 94.3 | 55.3 | 35.1 | 72.5 | 46.8 | 69.2 | **85.1** | 48.3 | 66.2 |
| CONCH | 74.7 | 97.9 | 59.4 | 63.2 | 78.7 | 58.3 | 73.5 | 79.8 | 33.0 | 68.7 |
| BiomedCLIP | 48.8 | 94.3 | 29.9 | 31.7 | 84.0 | 39.8 | 36.7 | 73.7 | 32.2 | 52.9 |
| PathGen-CLIP | **90.0** | 97.5 | 63.3 | 65.6 | **89.2** | 59.5 | 73.5 | 82.9 | 50.3 | 74.3 |
| PathGen-CLIP-L | 89.8 | **99.3** | **78.0** | **70.6** | 88.2 | **71.5** | **74.6** | 82.2 | **63.5** | **79.7** |

physicians assess each finding, with Pathologist A and Pathologist B assigning average accuracy ratings of 90.3% and 88.5% per caption, respectively. For comparison, we evaluate image-caption pairs from online platforms like Twitter, which demonstrate less than 80% accuracy in human assessments. Moreover, the number of findings per caption is significantly lower, as Twitter captions tend to be brief, less detailed, and often include irrelevant features such as aesthetic aspects rather than directly corresponding to the image. Consequently, the overall quality of PathGen-1.6M is substantially higher than that of existing datasets.

## 4 EXPERIMENTS

In this section, we describe the training process of PathGen-LLaVA$_{desp}$ and PathGen-CLIP utilizing our generated dataset. We then evaluate PathGen-CLIP's effectiveness through various downstream pathology tasks, comparing its performance against state-of-the-art baseline models. These evaluations encompass zero-shot image classification, few-shot image classification, and whole slide image classification. Finally, we demonstrate that by integrating the PathGen-CLIP vision encoder with LLMs and leveraging our dataset, we achieve superior performance in pathology-specific LMMs.

### 4.1 IMPLEMENT DETAILS OF MODEL TRAINING PROCESS

**PathGen-LLaVA$_{desp}$**: PathGen-LLaVA$_{desp}$ adopts LLaVA's model structure and training approaches, divided into two stages. Initially, we align PathGen-CLIP-L$_{init}$ with Vicuna LLM using image-text pairs from PathGen$_{init}$, facilitated by a fully connected (FC) layer. Subsequently, we fine-tune both the FC layer and the Vicuna component using pre-generated detailed image descriptions. This process equips PathGen-LLaVA$_{desp}$ with the capability to generate image descriptions.

**PathGen-CLIP:** The data generated using PathGen-LLaVA$_{desp}$ predominantly features extensive morphological descriptions. Additionally, the pretrained LLM component (Vicuna) has undergone human value alignment, often avoiding direct diagnostic outputs and frequently recommending consultation with a professional pathologist for definitive diagnoses. Therefore, we utilize PathGen-1.6M for first-stage training to help the model learn key morphological and tissue structural features. Subsequently, PathGen$_{init}$ is employed in the second stage of training, which enhances the model's diagnostic comprehension capabilities while building upon the strong morphological understanding established in first stage training. In the Appendix B.3, we conduct ablation studies to compare the effects of merging PathGen$_{init}$ and PathGen-1.6M for simultaneous training versus using PathGen-1.6M for the first stage of training followed by PathGen$_{init}$ in the second stage.

### 4.2 ZERO-SHOT IMAGE CLASSIFICATION

Due to the training of CLIP-based models on image-text pairs through contrastive learning, these models achieve an intrinsic alignment between textual descriptions and visual content. This alignment facilitates zero-shot image classification, which is particularly effective in scenarios with no annotations. To underscore the capabilities of the PathGen-CLIP series, we evaluate its zero-shot image classification performance on nine pathology classification datasets, including PatchCamelyon (Pcam) (Veeling et al., 2018), CRC-100K (Kather et al., 2018), SICAPv2 (Silva-Rodríguez et al., 2020), BACH (Aresta et al., 2019), Osteo (Arunachalam et al., 2019), SkinCancer (Kriegsmann

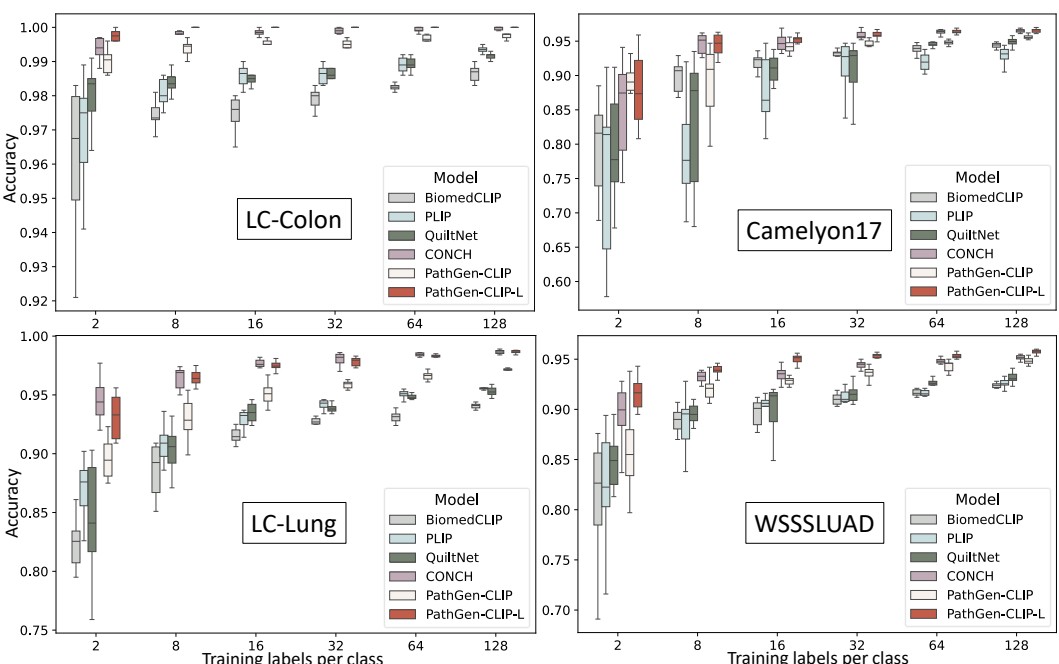

Figure 3: Comparison of few-shot classification accuracy (%) across different CLIP models on various pathology image classification datasets, visualized with box plots.

et al., 2022), WSSSLUAD (Han et al., 2022), LC-Lung, and LC-Colon (Borkowski et al., 2019). For each dataset, we design class-specific prompts, such as "an H&E image of class", and calculate the similarity between each class's text prompt and the image. The class prompt with the highest similarity score is assigned as the predicted label. We compare the performance of PathGen-CLIP with eight previous CLIP models, including OpenAI-CLIP, OpenAI-CLIP-L, PLIP, PMC-CLIP, PubMedCLIP, QuiltNet, PathCLIP, BiomedCLIP and CONCH.

***Results: The PathGen-CLIP series significantly outperforms previous SOTA models in zero-shot classification tasks, with PathGen-CLIP-L emerging as a particularly advanced model.*** As demonstrated in Table 2, PathGen-CLIP exceeds QuiltNet by 30.5% on the Pcam dataset and by 19.2% on the SkinCancer dataset. On average performance across all datasets, PathGen-CLIP also far surpasses the previously SOTA model, CONCH, by 6.4%. Moreover, our stronger variant, the PathGen-CLIP-L model, exhibits exceptionally consistent performance across various datasets, achieving remarkable results even in datasets where all other models perform poorly. For instance, on the BACH dataset, PathGen-CLIP-L already surpasses previous models by a large margin (13.2%). Similarly, it exceeds the average performance of PathGen-CLIP by 5.4%. The high performance of both PathGen and PathGen-CLIP-L underscores the effectiveness of our PathGen-1.6M dataset, offering potential for clinical utility in scenarios where no annotated data is available.

## 4.3 Few-shot Image Classification with Linear Probing

Traditional image classification tasks generally require extensive labeled data to achieve high accuracy, which is impractical in many real-world applications due to resource constraints, particularly in the pathology domain. In this study, we explore the effectiveness of PathGen-CLIP in a few-shot setting, where the model undergoes fine-tuning through linear probing on its pre-extracted feature representations. We assess the model on four representative datasets: LC-Colon, Camelyon17, LC-Lung, and WSSSLUAD, with fine-tuning performed using various training sizes (2, 8, 16, 32, 64 and 128 shots). Each training size is randomly sampled 10 times and conducts 10 runs. The results are displayed in box plots to illustrate the model's performance across different conditions.

***Results: The PathGen-CLIP series achieves notable performance with minimal samples, making the model viable in clinical settings with scarce annotations.*** As illustrated in Figure 3, with only 2 shots, PathGen-CLIP-L reaches close to 92% accuracy on the WSSSLUAD dataset, significantly

Table 3: Performance of various CLIP models across three datasets, two MIL methods, and two evaluation metrics. The top performance is highlighted in bold, with the second-best underlined. Additionally, the last row in gray indicates the performance of current state-of-the-art vision-only pre-trained pathology model.

| Performance / Method | | CAMELYON-17 | | CAMELYON-16 | | BRACS | | Average | |
|---|---|---|---|---|---|---|---|---|---|
| | | F1-score | AUC | F1-score | AUC | F1-score | AUC | F1-score | AUC |
| ABMIL | OpenAI-CLIP | $23.5_{\pm4.6}$ | $60.7_{\pm3.3}$ | $62.8_{\pm3.5}$ | $61.4_{\pm2.7}$ | $46.8_{\pm0.5}$ | $78.5_{\pm0.7}$ | 44.4 | 66.9 |
| | OpenAI-CLIP-L | $37.7_{\pm2.9}$ | $76.7_{\pm2.1}$ | $75.8_{\pm0.8}$ | $69.7_{\pm1.6}$ | $51.6_{\pm3.9}$ | $78.9_{\pm0.9}$ | 55.0 | 75.1 |
| | PLIP | $45.6_{\pm5.0}$ | $82.8_{\pm1.1}$ | $86.6_{\pm1.4}$ | $90.0_{\pm2.7}$ | $51.7_{\pm2.0}$ | $78.5_{\pm0.4}$ | 61.3 | 83.8 |
| | Quilt-Net | $44.3_{\pm2.0}$ | $84.2_{\pm1.0}$ | $82.9_{\pm1.6}$ | $87.1_{\pm2.2}$ | $54.7_{\pm3.0}$ | $82.3_{\pm1.6}$ | 60.6 | 84.5 |
| | BiomedCLIP | $55.5_{\pm2.5}$ | $84.1_{\pm1.2}$ | $82.6_{\pm1.4}$ | $83.6_{\pm3.4}$ | $57.4_{\pm4.8}$ | $80.6_{\pm1.4}$ | 65.2 | 82.8 |
| | PathCLIP | $45.2_{\pm3.3}$ | $82.6_{\pm1.9}$ | $80.2_{\pm1.5}$ | $85.0_{\pm1.5}$ | $56.4_{\pm3.2}$ | $83.8_{\pm0.8}$ | 60.6 | 83.8 |
| | CONCH | $55.0_{\pm2.0}$ | $86.4_{\pm0.7}$ | $93.9_{\pm1.1}$ | $95.2_{\pm0.7}$ | $62.0_{\pm1.4}$ | $\mathbf{90.0}_{\pm0.6}$ | 70.3 | $\mathbf{90.5}$ |
| | PathGen-CLIP (ours) | $\underline{58.2}_{\pm3.3}$ | $87.5_{\pm1.1}$ | $93.5_{\pm2.2}$ | $\mathbf{96.9}_{\pm1.5}$ | $62.6_{\pm1.9}$ | $85.8_{\pm0.8}$ | $\underline{71.4}$ | 90.1 |
| | PathGen-CLIP-L (ours) | $\mathbf{58.6}_{\pm6.7}$ | $\mathbf{87.9}_{\pm1.2}$ | $\mathbf{94.3}_{\pm1.8}$ | $95.8_{\pm1.4}$ | $\mathbf{66.6}_{\pm6.4}$ | $87.2_{\pm2.9}$ | $\mathbf{73.2}$ | $\underline{90.3}$ |
| | GigaPath-G | $54.4_{\pm3.9}$ | $88.3_{\pm1.2}$ | $98.0_{\pm0.4}$ | $98.3_{\pm0.2}$ | $62.3_{\pm4.9}$ | $84.2_{\pm1.1}$ | 71.6 | 90.2 |
| ACMIL | OpenAI-CLIP | $25.4_{\pm4.1}$ | $59.4_{\pm3.7}$ | $63.7_{\pm4.3}$ | $67.3_{\pm4.3}$ | $52.2_{\pm4.5}$ | $76.7_{\pm1.6}$ | 47.1 | 67.8 |
| | OpenAI-CLIP-L | $34.9_{\pm4.5}$ | $78.5_{\pm3.0}$ | $78.7_{\pm3.7}$ | $73.4_{\pm3.4}$ | $55.5_{\pm3.6}$ | $78.8_{\pm2.0}$ | 56.4 | 76.9 |
| | PLIP | $46.0_{\pm1.5}$ | $86.1_{\pm1.0}$ | $90.4_{\pm2.2}$ | $94.8_{\pm1.3}$ | $57.0_{\pm3.3}$ | $80.8_{\pm0.7}$ | 64.5 | 87.2 |
| | Quilt-Net | $44.4_{\pm1.0}$ | $86.0_{\pm0.9}$ | $84.8_{\pm3.5}$ | $90.5_{\pm3.4}$ | $60.8_{\pm3.5}$ | $82.0_{\pm1.9}$ | 63.3 | 86.2 |
| | BiomedCLIP | $\underline{53.6}_{\pm4.0}$ | $83.6_{\pm1.3}$ | $82.9_{\pm2.0}$ | $84.8_{\pm3.3}$ | $63.2_{\pm2.0}$ | $81.9_{\pm0.8}$ | 66.6 | 83.4 |
| | PathCLIP | $44.2_{\pm0.6}$ | $82.9_{\pm1.3}$ | $84.4_{\pm1.6}$ | $87.3_{\pm1.6}$ | $58.0_{\pm7.6}$ | $83.0_{\pm2.4}$ | 62.2 | 84.4 |
| | CONCH | $56.3_{\pm3.6}$ | $87.5_{\pm0.9}$ | $94.4_{\pm1.0}$ | $97.2_{\pm0.5}$ | $\underline{66.1}_{\pm1.6}$ | $\mathbf{88.7}_{\pm0.8}$ | $\underline{72.7}$ | 91.1 |
| | PathGen-CLIP (ours) | $53.3_{\pm4.6}$ | $89.4_{\pm1.2}$ | $92.6_{\pm1.6}$ | $97.2_{\pm0.9}$ | $\mathbf{66.9}_{\pm3.0}$ | $87.0_{\pm0.4}$ | 71.0 | $\underline{91.2}$ |
| | PathGen-CLIP-L (ours) | $\mathbf{58.4}_{\pm5.2}$ | $\mathbf{92.0}_{\pm0.7}$ | $\mathbf{94.5}_{\pm1.0}$ | $\mathbf{97.4}_{\pm1.9}$ | $\mathbf{66.9}_{\pm5.0}$ | $88.4_{\pm1.4}$ | $\mathbf{73.3}$ | $\mathbf{92.6}$ |
| | GigaPath-G | $55.9_{\pm3.4}$ | $89.8_{\pm1.4}$ | $95.7_{\pm1.2}$ | $99.2_{\pm0.4}$ | $61.0_{\pm4.5}$ | $83.6_{\pm1.1}$ | 70.9 | 90.9 |

outperforming competitors like PLIP, Quilt-NET, BiomedCLIP and CONCH which record accuracies below 90%. Furthermore, as the number of shots increases, PathGen demonstrates faster performance improvements and convergence compared to other models. Additionally, the narrower box plot widths of the PathGen-CLIP model highlight its superior task adaptation capabilities and robustness.

## 4.4 WHOLE SLIDE IMAGE CLASSIFICATION

Whole slide image classification is essential for automating disease identification and classification from high-resolution pathological slides by analyzing high-resolution pathological slide images, which are typically larger than $100{,}000 \times 100{,}000$ pixels. This task is particularly valuable for clinical practitioners as it boosts clinical accuracy and efficiency significantly. The standard approach for WSI classification involves segmenting WSIs into image patches, extracting instance embeddings using a frozen image encoder, and employing Multiple Instance Learning (MIL) to convert these embeddings into slide-level predictions. Superior patch representations provided by the image encoder are crucial, as they significantly influence WSI prediction performance. Therefore, we assess the efficacy of the PathGen-CLIP series in comparison with other prominent models, including OpenAI-CLIP, OpenAI-CLIP-L, PLIP, BiomedCLIP, PathCLIP, QuiltNet and CONCH. For the MIL method, we utilize the widely adopted ABMIL (Ilse et al., 2018) and the current SOTA method, ACMIL (Zhang et al., 2023b). Additionally, we involve the state-of-art vision-only encoder, GigaPath-G (Xu et al., 2024), which is four times larger and trained on 800 times the amount of vision-only private data for comparison. Our evaluations span three datasets: CAMELYON16 (Litjens et al., 2018), CAMELYON17 (Litjens et al., 2018), and BRACS (Brancati et al., 2022), excluding any TCGA-related datasets. For detailed experimental setup, please refer to the Appendix C.3.

***Results: PathGen-CLIP series consistently outperform existing pathology-specific CLIP models Across three key datasets leveraging both MIL methods.*** For instance, as shown in Table 3, the PathGen-CLIP model, employing the ABMIL architecture, achieves a remarkable average AUC of 96.9 on the CAMELYON16 dataset. This substantially exceeds PLIP (90.01), BiomedCLIP (83.6), Quilt-Net (87.1) and CONCH (95.2). Overall, PathGen-CLIP-L models demonstrate significant improvements with an average AUC of 92.6% across datasets using ACMIL—outperforming PLIP at 87.2%, BiomedCLIP at 83.4%, Quilt-Net at 86.2%, and CONCH at 91.1%. Despite CONCH being a powerful model trained on proprietary datasets, our PathGen-CLIP series—trained on PathGen-1.6M, an open-source variant—significantly outperforms these established models.

Table 4: Overall results of models on the PathMMU **test set**. The best-performing LMM in each subset for general and pathology domain LMMs is **in-bold**, and the top-performing LMM is underlined.

| | Test Overall | | PubMed | | SocialPath | | EduContent | | Atlas | | PathCLS | |
|---|---|---|---|---|---|---|---|---|---|---|---|---|
| | Tiny (1156) | ALL (9677) | Tiny (281) | ALL (3068) | Tiny (235) | All (1855) | Tiny (255) | All (1938) | Tiny (208) | ALL (1007) | Tiny (177) | ALL (1809) |
| Expert performance | 71.8 | - | 72.9 | - | 71.5 | - | 69.0 | - | 68.3 | - | 78.9 | - |
| **General Large Multimodal Models** | | | | | | | | | | | | |
| BLIP-2 FLAN-T5-XXL | 33.3 | 33.5 | 37.0 | 37.4 | 35.7 | 34.6 | 30.2 | 34.5 | 39.4 | 40.7 | 19.8 | 20.6 |
| InstructBLIP-FLAN-T5-XXL | 34.3 | 33.9 | 39.1 | 37.2 | 33.6 | 34.3 | 34.5 | 36.0 | 38.5 | 39.3 | 22.6 | 22.7 |
| LLaVA-1.5-13B | 38.8 | 37.6 | 44.5 | 41.0 | 40.4 | 40.4 | 34.1 | 39.4 | 47.1 | 44.3 | 24.9 | 23.5 |
| Qwen-VL-MAX | 49.2 | 45.9 | 53.0 | 50.9 | 53.6 | 49.3 | 52.2 | 47.9 | 51.4 | 49.8 | 30.5 | 29.6 |
| Gemini Pro Vision | 42.8 | 42.7 | 43.8 | 44.9 | 42.4 | 42.0 | 43.5 | 43.7 | 49.5 | 49.4 | 32.8 | 34.7 |
| GPT-4V-1106 | 53.9 | 49.8 | 59.4 | 53.5 | 58.7 | 53.9 | 60.4 | 53.6 | 48.1 | 52.8 | 36.2 | 33.8 |
| **Pathology-specific Large Multimodal Models** | | | | | | | | | | | | |
| LLaVA-Med | 25.3 | 26.2 | 28.5 | 27.7 | 28.9 | 27.3 | 22.7 | 27.2 | 22.6 | 30.7 | 22.6 | 20.3 |
| Quilt-LLaVA | 45.6 | 41.5 | 47.3 | 42.6 | 46.4 | 46.6 | 51.8 | 45.3 | 46.2 | 42.7 | 32.2 | 29.2 |
| PathGen-LLaVA | **60.1** | **58.4** | **60.1** | **60.1** | **60.9** | **58.8** | **60.8** | **60.7** | **63.5** | **64.9** | **54.2** | **48.9** |

***Stronger models have the potential to be trained with fewer data through a vision-language approach, compared to traditional vision-only pretraining.*** Notably, PathGen-CLIP-L outperforms GigaPath-G on WSI classification tasks, achieving a 2.4% higher F1-score and 1.7 AUC improvement with ACMIL, despite GigaPath-G having significantly more model parameters and training data. This underscores the substantial advancements of PathGen-1.6M contributes to the analysis of WSIs.

## 4.5 INTEGRATING WITH LARGE LANGUAGE MODELS

LLMs possess extensive knowledge and common sense due to their larger model sizes and diverse training datasets. Models like CLIP, which are already aligned with language models, are often used for integration with LLMs to develop powerful LMMs. In this work, to train such LMMs, we construct 200K instruction-tuning samples based on PathGen data by prompting GPT-4 using the prompt shown in Figure 16 and Figure 17 in the appendix to generate instruction-tuning data, including 95K multi-choice QAs and 105K multi-round of dialogues. We follow the training methodology of LLaVA to train our LMM, which we refer to as PathGen-LLaVA. We evaluate the performance of PathGen-LLaVA on the PathMMU dataset, which includes expert annotations from multiple sources and diseases. Additionally, we compare PathGen-LLaVA with the most advanced general-domain models, such as GPT-4V, Gemini-Pro Vision, Qwen-VL-Max, as well as previous domain-specific LMMs, including LLaVA-Med and Quilt-LLaVA, to validate the capabilities of our model. For detailed information regarding training data and model training details, please refer to the Appendix C.5.

***Results: PathGen-LLaVA significantly outperforms previous SOTA pathology LMMs, even surpassing the leading general model, GPT-4V.*** As shown in Table 4, PathGen-LLaVA consistently exceeded Quilt-LLaVA by 17.5%, 12.2%, 15.4%, and 22.2% across the PathMMU's PubMed, SocialPath, Atlas, and PathCLS subsets respectively, and also outperformed the top general-domain model GPT-4V by a significant margin. Specifically, in overall test performance, PathGen-LLaVA achieved 58.4%, surpassing GPT-4V's 49.8%. These results demonstrate the superiority of the PathGen dataset and the effectiveness of PathGen-CLIP as a backbone. It marks a step closer to the potential application of pathology LMMs in assisting medical professionals in practice.

## 4.6 EXPLORATION OF SCALABILITY AND VERSATILITY OF PATHGEN-1.6M

**Scaling up with Non-WSI Report Paired Data:** In our approach, we require WSIs paired with corresponding reports. However, many WSIs in datasets like TCGA lack associated reports. To explore whether expanding our dataset without paired reports can enhance performance. We conduct experiments by adding 6,800 additional WSIs from TCGA that lack associated reports and extract a total of 1,238,750 patches using source prompts (247,750) and k-means clustering (991,000). These newly collected samples, derived from source prompts and k-means clustering, were then incorporated into the PathGen-1.6M dataset to train the CLIP model, referred to as PathGen-CLIP-L*.

***Results: Scaling up with non-WSI report paired data using PathGen's generation approach and incorporating it into PathGen-1.6M still improves zero-shot performance on 7 out of 9 tasks*** (as shown in Table 5), confirming the scalability of our method regardless of WSI report availability.

Table 5: Results of scaling up data (PathGen-CLIP-L*), scaling up the model (PathGen-CLIP-H), and utilizing PathGen-1.6M to adapt vision-only model to vision-language model (PathGen-GigaPath-G).

| Method | LC-Lung | LC-Colon | CRC-100K | SkinCancer | Pcam | BACH | Osteo | WSSSLUAD | SICAPv2 |
|--------|---------|----------|----------|------------|------|------|-------|----------|---------|
| PathGen-CLIP-L | 90.0 | 99.3 | 78.0 | 70.6 | 88.2 | 71.5 | 74.6 | 82.9 | 63.5 |
| PathGen-CLIP-L* | 92.5 | 99.6 | 78.1 | 72.1 | 89.9 | 67.9 | 77.5 | 84.4 | 59.5 |
| PathGen-CLIP-H | 94.4 | 98.5 | 78.1 | 76.4 | 89.4 | 66.5 | 81.4 | 85.3 | 56.5 |
| PathGen-GigaPath-G | 94.6 | 98.1 | 75.0 | 74.2 | 89.0 | 58.5 | 80.2 | 84.5 | 46.6 |

Table 6: Comparison of GigaPath-G and PathGen-GigaPath-G for the few-shot classification task using the Camelyon17 and WSSSLUAD datasets. The top accuracy is highlighted in **bold**.

| Datasets | Models | Number of Shots | | | |
|----------|--------|-----------------|---|---|---|
| | | 1 | 4 | 16 | 64 |
| Camelyon17 | GigaPath-G | 60.30% ± 13.55% | 82.93% ± 15.10% | **94.24% ± 4.15%** | 95.83% ± 3.30% |
| | PathGen-GigaPath-G | **72.71% ± 23.35%** | **84.11% ± 13.95%** | 94.02% ± 6.25% | **97.40% ± 0.75%** |
| WSSSLUAD | GigaPath-G | 67.76% ± 12.55% | 76.55% ± 9.55% | 87.76% ± 5.10% | 93.33% ± 1.15% |
| | PathGen-GigaPath-G | **77.96% ± 23.30%** | **91.06% ± 3.35%** | **93.79% ± 1.20%** | **94.82% ± 0.50%** |

**Scaling up with Larger Model:** In an era where researchers continually scale up models to achieve greater capabilities, the adaptability of data for larger models is crucial. Therefore, we conduct further experiments by scaling up PathGen-CLIP-L to PathGen-CLIP-H to validate that our data remains effective for larger models.

*Results: PathGen-CLIP-H achieves better results than PathGen-CLIP-L on 6 out of 9 zero-shot classification datasets*, as shown in Table 5. This clearly demonstrates that PathGen-1.6M effectively supports larger models in achieving enhanced performance.

**Incorporation with Advanced Vision-only Model:** To demonstrate the versatility of PathGen-1.6M, we transition from vision-language pretrained models to the state-of-the-art vision-only pathology pretrained encoder, GigaPath-G (Xu et al., 2024), trained on over 1 billion private patch images. We combine it with the language encoder BioClinical-BERT (Alsentzer et al., 2019) to create a new CLIP model, PathGen-GigaPath-G.

*Results: PathGen-GigaPath-G not only acquires vision-language capabilities but also enhances its vision-only performance.* As shown in Table 5, PathGen-GigaPath-G achieves comparable overall performance on zero-shot tasks, despite not being pre-trained on billion-scale datasets like OpenAI-CLIP. Furthermore, our linear probing evaluations on two datasets, presented in Table 6, reveal substantial performance improvements, particularly with a very limited number of samples. These results indicate that PathGen-1.6M is not only effective for training CLIP-based models but also suitable for vision-only models, underscoring its substantial potential impact.

## 5 CONCLUSION

In this work, we propose a novel multi-agent collaboration approach that generates 1.6 million high-quality pathology image-text pairs from whole slide images. Utilizing these generated data alongside existing datasets, we train two advanced models: PathGen-CLIP and PathGen-CLIP-L. These models achieve significant advancements in zero-shot image classification, few-shot image classification, and whole slide image classification, even attaining comparable or better results to much larger vision-only models in terms of parameters and training data. Additionally, We integrate PathGen-CLIP-L with an LLM to create a powerful pathology-specific LMM, PathGen-LLaVA. By leveraging the PathGen-1.6M dataset, we generate 200,000 instruction-tuning samples to train PathGen-LLaVA. Our experiments demonstrate that PathGen-LLaVA exhibits strong pathology image understanding capabilities, significantly outperforming previous pathology LMMs on the large-scale PathMMU dataset by a large margin and surpassing the performance of powerful closed-source GPT-4V. Furthermore, we investigate the scalability of PathGen-1.6M by scaling up the training data, scaling up the model size, and combining it with advanced vision-only pre-trained pathology models, showcasing the extensibility of the PathGen approach. Extensive evaluations confirm the superiority and promising potential of the PathGen-1.6M dataset.

## 6 ETHICS STATEMENT

Our research is solely based on image-text pairs, with no involvement of human subjects. The dataset used in this study is publicly released and is maintained for long-term availability on GitHub. The data consists of descriptive content related to pathology images, without any inclusion of potentially harmful insights or conclusions. There are no potential conflicts of interest or sponsorship associated with this research. The PathGen dataset utilized in this study is derived from publicly available data provided by TCGA, ensuring full compliance with privacy, security, and legal requirements. All relevant ethical standards and research integrity guidelines have been thoroughly observed throughout the development and use of this dataset.

## 7 REPRODUCIBILITY STATEMENT

The datasets proposed in this study are publicly released and maintained for long-term accessibility at GitHub PathGen-1.6M. Additionally, we have made PathGen-CLIP, PathGen-CLIP-L, and PathGen-LLaVA openly available, enabling the research community to fully reproduce our results. Furthermore, detailed parameter settings for all experiments and comprehensive descriptions of downstream datasets are provided in the appendix. These resources support the verification and extension of our work by other researchers.

## 8 ACKNOWLEDGEMENTS

This study was partially supported by Zhejiang Provincial Natural Science Foundation of China (Grant No.XHD23F0201), the National Natural Science Foundation of China (Grant No.92270108), foundation of Muyuan Laboratory (Program ID: 14106022401,14106022402), the Research Center for Industries of the Future (RCIF) at Westlake University, the Westlake University-Muyuan Joint Research Institute (Program ID: WU2022MY003) and the Westlake Education Foundation.

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

# A  OVERVIEW OF PATHGEN-1.6M

In this section, we present details of the PathGen-1.6M dataset, including quality assessment and statistical information, to illustrate its high quality and significant contribution to the open-source pathology dataset community. Additionally, we provide further details about the derivative dataset, PathGen-Instruct-200K, introduced in Section 4.5 and specifically designed for training large multi-modal models. Finally, we outline the procedure for accessing our datasets.

## A.1  IMAGE QUALITY

As mentioned in the main paper, the quality of previous pathology datasets is significantly degraded due to screenshots and various compression issues. We conduct a comparative image quality analysis between PathGen and previous datasets—PathCap (Sun et al., 2024b), Quilt (Ikezogwo et al., 2024), and OpenPath (Huang et al., 2023). We randomly sample 500 images from each dataset. These images are assessed using the existing image clarity model[1] and through human evaluations conducted by pathologists.

In evaluating image clarity with clarity assessment models, the field of view and size significantly affect scores. To standardize assessments, we implement two approaches: (1) cropping the original images to a uniform size of $672 \times 672$, and (2) resizing the original images to $336 \times 336$, which is the input image size for PathGen-CLIP-L. For the pathologists' human evaluation, we present the original images without resizing or cropping. The pathologists are tasked with categorizing the image quality into five levels: low, moderately low, medium, moderately high, and high, corresponding to scores of 1-5.

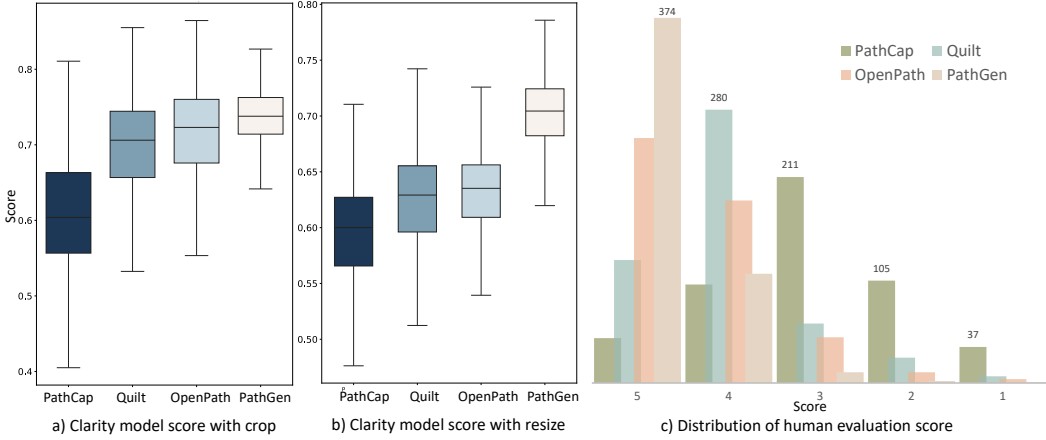

a) Clarity model score with crop  b) Clarity model score with resize  c) Distribution of human evaluation score

Figure 4: Comparison of image quality with the previous model using the clarity assessment model and human evaluation.

***Results: The image quality of PathGen is notably superior to other datasets, significantly higher than those from PathCap, Quilt, and OpenPath.*** As illustrated in the left and middle sections of Figure 4, we display the clarity scores of each dataset's images in a boxplot format. The overall image quality from PathGen is significantly higher than those from earlier datasets, especially after resizing to $336 \times 336$. Additionally, the shorter length of the box for PathGen images indicates more consistent image quality. This consistency is likely due to the images being directly cropped from WSIs with minimal compression and distortion. Similarly, as shown in the right part of Figure 4, the human evaluation experiments further confirm that PathGen consistently delivers higher image quality scores. This demonstrates that our image quality more closely approximates the images used by clinicians during diagnostic readings. This is highly significant in reducing the domain gap between training images and those used in clinical scenarios.

---

[1]https://help.aliyun.com/zh/viapi/use-cases/image-clarity-grade-1

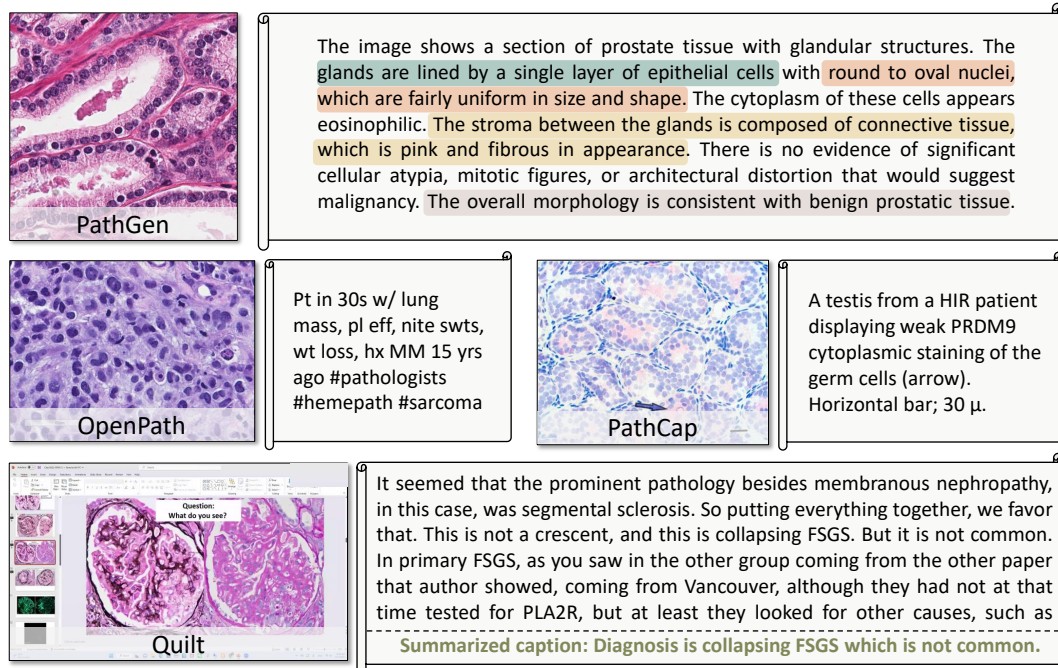

Figure 5: Comparison of caption quality with previous datasets, with different pathological descriptions from various aspects of PathGen marked in different background colors.

## A.2 CAPTION QUALITY

There are several issues with captions from existing datasets: **(1) Brevity:** Captions are often too brief to comprehensively outline the details within an image. This lack of detail can cause models to overlook important features in the image. This issue is prevalent in datasets like OpenPath, PathCap, and Quilt. **(2) Mismatch between text and image:** On social media platforms like Twitter, the text accompanying an image may not necessarily describe the pathological features of the image but might focus on aesthetic qualities instead. On YouTube, a current frame might describe content from previous frames, leading to mismatches. **(3) Colloquial language:** Captions are often informal and random on social platforms. In contrast, the PathGen-1.6M dataset employs captions generated by a pathology-specific LMM, which are further refined by various agents to ensure relevance between the text and the image and detailed descriptions. As illustrated in Figure 5, it is evident that the captions in PathGen are significantly more detailed and precise. It recognizes cell morphology ("round to oval nuclei, fairly uniform in size and shape"), tissue structure ("The glands are lined by a single layer of epithelial cells"), and diagnosis ("overall morphology is consistent with benign prostatic tissue"). In contrast, OpenPath's caption contains many abbreviations, which are not in the vocabulary of models like BERT (Devlin et al., 2019) and thus are identified as '[UNK]' tokens, leading to the loss of important information. PathCap only includes a single piece of staining information, while Quilt, despite containing a lengthy text, mostly discusses irrelevant details. The effective description related to the image boils down to only one sentence.

## A.3 DATASET STATISTICS

**Image Sources:** PathGen sources its image data from TCGA (The Cancer Genome Atlas), which provides a comprehensive collection of pathology whole-slide images from various participating institutions, encompassing a diverse range of tissue and cancer types. We illustrate the data distribution of different image sources in the left panel of Figure 6. Our dataset originates from 27 different institutions, with the highest number of images obtained from breast tissue, totaling 217,025 images, and the fewest from lymphoid tissue, containing 4,543 images.

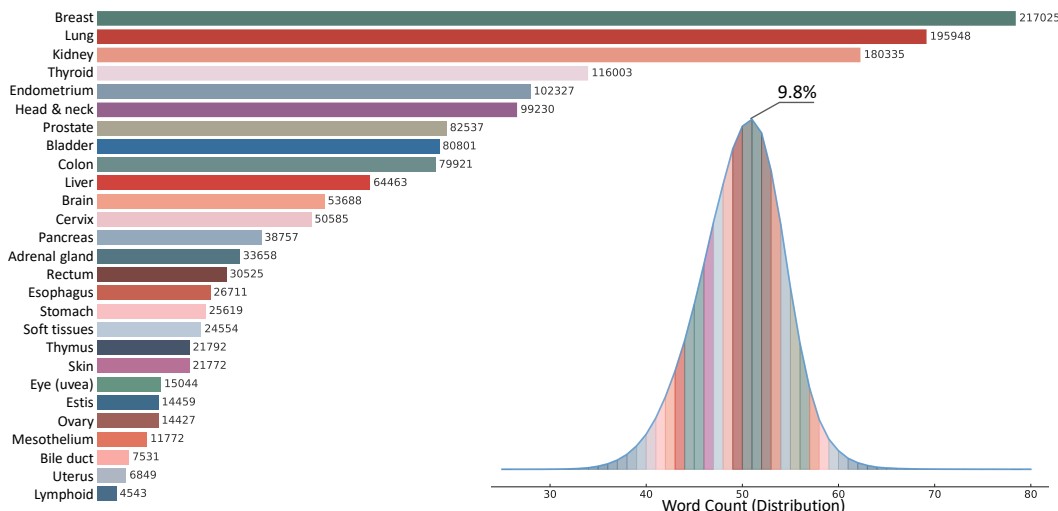

Figure 6: Visualization of the distribution of sample quantities from different tissue sources (top right) and the distribution of caption lengths (bottom right).

**Word Count:** We conduct a distribution analysis of the caption lengths for all samples, as shown in the lower right corner of Figure 6. The distribution of caption lengths is relatively symmetrical. Captions shorter than 30 are extremely rare, indicating that our captions tend to describe the details of pathological images. The largest proportion of captions is around 50 words, accounting for 9.8%, while captions exceeding 70 words are similarly uncommon. This is because we employ a summary agent to control the length, ensuring compatibility with CLIP's maximum input length of 77 tokens.

**Word Frequency:** We also analyze the word frequency distribution within our dataset using word clouds, as shown in the upper left corner of Figure 7. Overall, the captions in PathGen heavily focus on morphological features such as "disorganized," "disrupted," "fibrous," "prominent," and diagnostic characteristics like "inflammatory," "neoplastic," and "atypia." The other three panels in Figure 7 display word frequency analyses for samples drawn from three distinct tissue types. Each tissue type exhibits a distinct word frequency distribution, closely related to the morphological and diagnostic features typical of that tissue, such as "alveolar" in lung tissue and "colloid" in thyroid tissue. This diversity in samples and captions enables our model to learn a broader range of feature knowledge.

A.4   STATISTICS OF DERIVATIVE DATASET PATHGEN-INSTRUCT-200K

As mentioned in the main text, to integrate with large language models and construct a large multimodal model, we extract samples from PathGen-1.6M to build PathGen-Instruct-200K. This dataset includes data for multi-choice QA-based instruction-tuning and dialogue-based open-ended instruction-tuning. We present the average question length, option length, answer length, and the number of questions with a single image for these two types of data, as shown in Table 7.

Table 7: Statistical Information for PathGen-Instruct-200K.

| Source | Avg. question length | Avg. option length | Avg. answer length | # Questions / # Images |
|---|---|---|---|---|
| Multi-choice QA | 14.9 | 3.7 | - | 1.2 |
| Open-ended QA | 17.1 | - | 33.2 | 5.0 |

B   ADDITIONAL EXPERIMENTS AND DISCUSSION ABOUT THE PATHGEN-1.6M

B.1   COMPARISON BETWEEN PATHGEN-LLAVA$_{desp}$, LLAVA-MED-V1.5, AND QUILT-LLAVA

Since the quality of description generation largely determines the effectiveness of PathGen, we compare PathGen-LLaVA$_{desp}$ with previous pathology LMMs, including Quilt-LLaVA (Ikezogwo

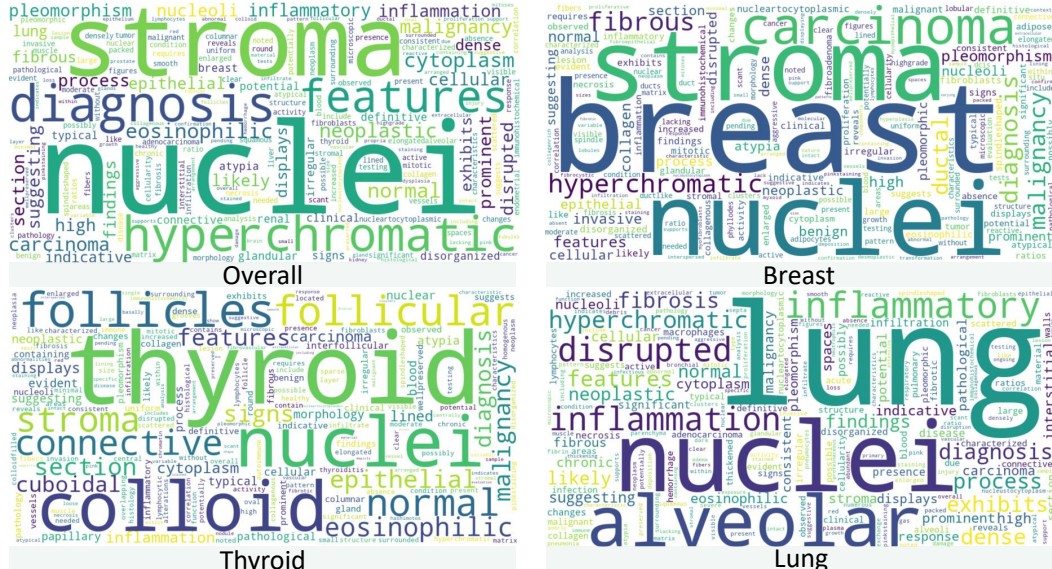

Figure 7: Visualization of word clouds derived from the overall dataset captions, along with separate word clouds for three distinct tissue sources.

et al., 2024) and LLaVA-Med-v1.5 (Li et al., 2023a), in terms of description generation. Our evaluation includes extensive testing with the state-of-the-art LMM GPT-4o (OpenAI, 2024), as well as a smaller-scale human evaluation.

For the GPT-4o testing, we randomly select 1000 images from our complete set of images, using GPT-4o as the evaluator to judge the quality of descriptions generated by both models. Previous research (Wang et al., 2023a) reveals that earlier versions of GPT-4 (OpenAI, 2023a) and Chat-GPT (OpenAI, 2022) exhibit position bias, where the model presented first is often judged more favorably. Since we are unsure if the latest versions of GPT-4 and GPT-4o still exhibit this behavior, we conduct two rounds of testing to ensure fairness. In one round, the description generated by PathGen-LLaVA$_{desp}$ is presented first, and in the other round, the compared model's description is presented first.

For the human evaluation, due to the high cost and difficulty of involving professional pathologists, we randomly select 100 images from our dataset and invite a professional pathologist to evaluate which model performs better.

***Results: The quality of descriptions generated by PathGen-LLaVA$_{desp}$ significantly surpasses that of LLaVA-Med-v1.5 and Quilt-LLaVA in both GPT-4o and human evaluations.*** As depicted in Figure 8, in the GPT-4o evaluation, PathGen-LLaVA$_{desp}$ outperforms Quilt-LLaVA in over 85% of cases and exceeds LLaVA-Med-v1.5 in more than 94% of cases. In the human evaluation, PathGen-LLaVA$_{desp}$ outperforms Quilt-LLaVA and LLaVA-Med-v1.5 in 85% and 98% of cases, respectively. Note that this comparison uses PathGen-LLaVA$_{init}$, an initial version of PathGen-LLaVA, to validate the quality of the dataset's descriptions. The performance gap might be more substantial if the more advanced PathGen-LLaVA is used.

Interestingly, in the GPT-4o evaluation, there appears to be a slight preference for the model whose response is positioned second. This bias is observed in both comparisons, whether PathGen-LLaVA$_{desp}$ is compared to Quilt-LLaVA or LLaVA-Med-v1.5. When the position is reversed (i.e., the comparison model's response is presented first and our model's response second), the winning rate of our model is slightly higher, although the difference is not substantial. We hypothesize that OpenAI may have optimized this position bias issue.

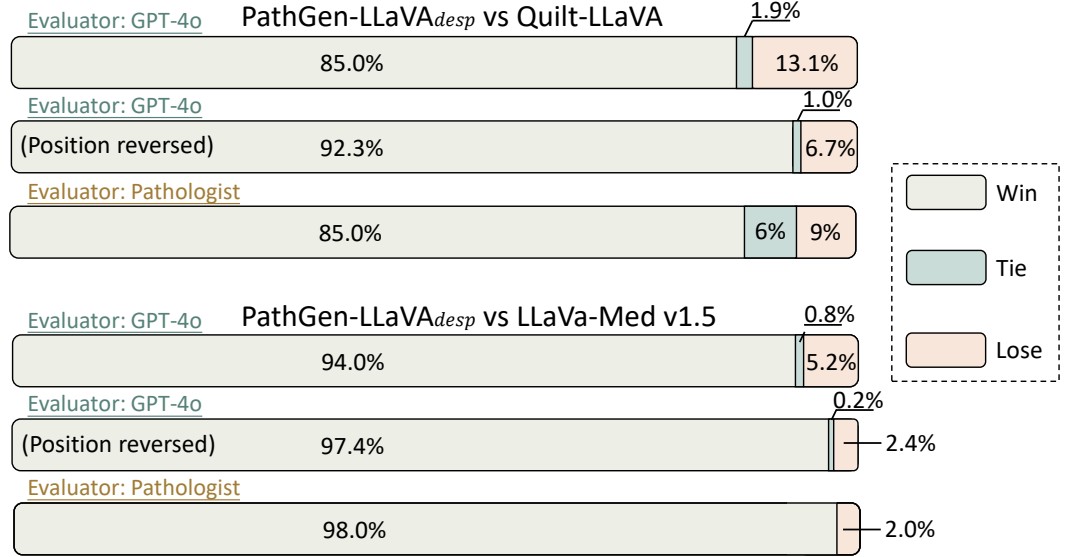

Figure 8: Comparison of performance in generating pathology image descriptions among PathGen-LLaVA$_{desp}$, Quilt-LLaVA, and LLaVA-Med-v1.5, evaluated through GPT-4o and human assessments.

## B.2 Performance Comparison across Different Zero-shot Classification Prompts

Existing medical or pathology-specific CLIP models usually design prompts that are particularly suited to their models to enhance zero-shot image classification capabilities. However, this specificity can skew direct comparisons between models due to prompt variability. To address this, we collect a substantial variety of prompts from existing literature on pathology-specific CLIP models and conduct a comparative test across eight mainstream datasets: LC-Lung, LC-Colon, Pcam, WSSSLUAD, CRC-100K, SkinCancer, BACH, and Osteo. We visualize performance variations using box plots to highlight the fluctuations in model responses to different prompts.

***Results: PathGen-CLIP and PathGen-CLIP-L consistently outperform previous models across all datasets and various prompts.*** As illustrated in Figure 9, the box plots for both models are significantly higher than those of other models. Additionally, the shorter box lengths indicate greater robustness of PathGen-CLIP and PathGen-CLIP-L to prompt variations. This superior performance and robustness make our models more suitable for practical clinical applications.

## B.3 Why PathGen-1.6M is Recommend for First-stage Training?

As we mentioned in the main paper, the captions generated by PathGen-LLaVA$_{desp}$ tend to describe morphological content, such as cellular and tissue structures, rather than direct diagnoses. This is due to the value alignment in the LLM, which avoids making definitive pathological diagnoses. Additionally, the training dataset for multimodal description generation contains relatively few diagnostic data entries. Therefore, using PathGen alone is insufficient for equipping PathGen-CLIP with diagnostic capabilities. In our study, we conducted experiments comparing the training of models using various scales of data solely from PathGen, as well as combining data from different scales of PathGen with PathGen$_{init}$.

***Results: We observe a notable decline in the performance of PathGen-CLIP across four datasets when the data scale exceeds 0.7 million.*** As depicted in Figure 10, this trend is consistent on most datasets whether the training utilizes data exclusively from PathGen or a combination of PathGen and PathGen$_{init}$ datasets. We hypothesize that when the scale of PathGen data surpasses that of PathGen$_{init}$, the model's excessive focus on morphological features of cells and tissues detracts from its ability to make direct pathological assessments. This shift in focus adversely affects the

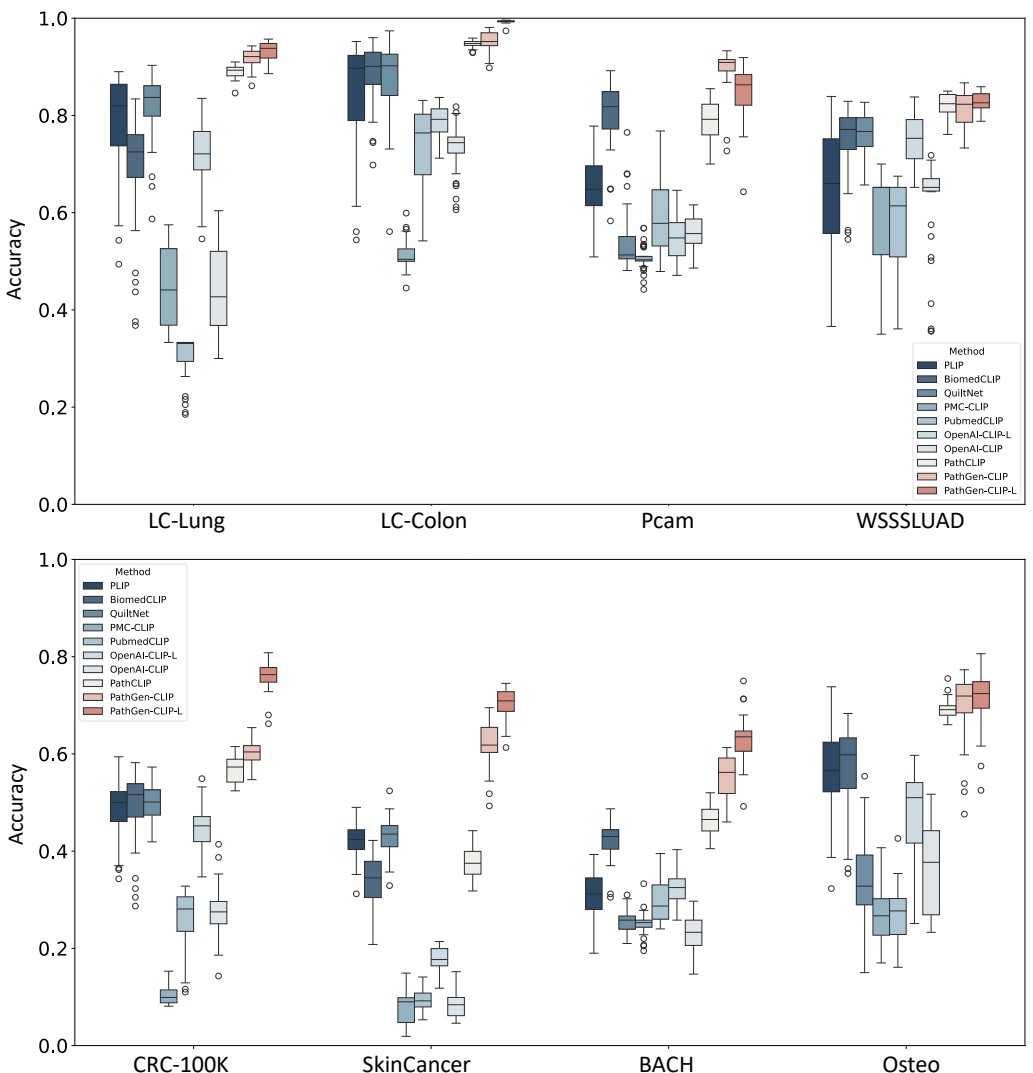

Figure 9: Boxplot visualization of performance variation between PathGen-CLIP and previous related CLIP models across various zero-shot classification prompts.

model's capability for zero-shot image classification, as it becomes overly specialized in recognizing structural details at the expense of broader diagnostic accuracy.

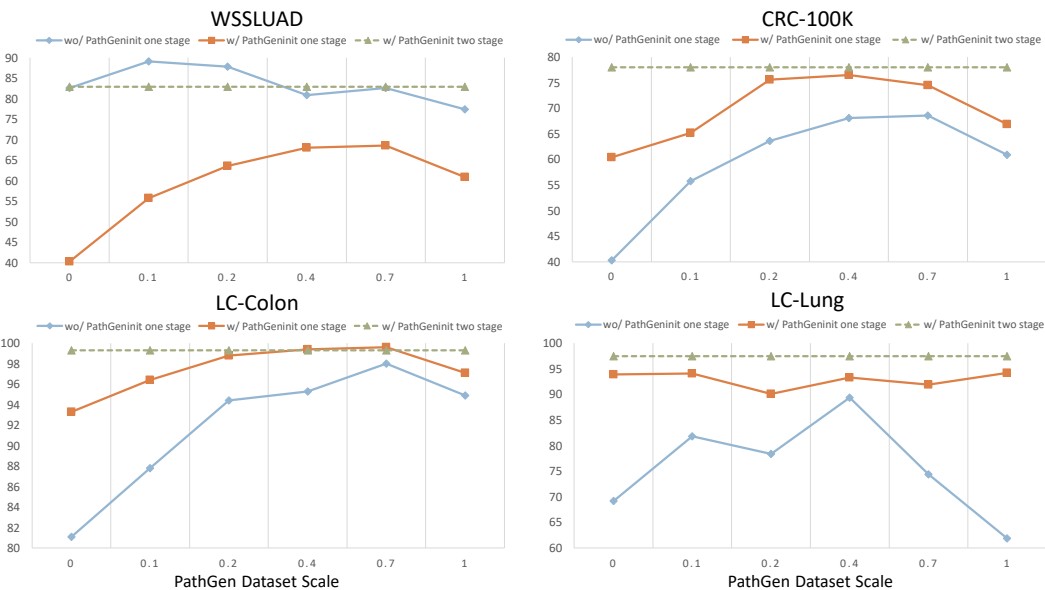

Figure 10: Illustration of different data construction strategies and comparisons of data scales. "wo/ PathGen$_{init}$ one stage" represents training solely with various scales of PathGen data using a single-stage training approach. "w/ PathGen$_{init}$ one stage" denotes the combination of different scales of PathGen and PathGen$_{init}$ data for single-stage training, a scale of 0 means all the training data is from PathGen$_{init}$. "w/ PathGen$_{init}$ two-stage" serves as a reference constant line, representing the final result of our current method, which employs PathGen-1.6M data for the first stage of training and then utilizes PathGen$_{init}$ for the second stage. The horizontal axis represents the data scale in millions.

## B.4 Performance Comparison Between Zero-Shot and Fully Supervised Approaches

To demonstrate the extent of the gap between zero-shot classification and full fine-tuning in the field of pathology, we conduct a comparative analysis of their results. As shown in Table 8, there remains a notable gap between zero-shot performance and full fine-tuning. However, the advancements of PathGen-CLIP-L significantly narrow this gap, showcasing its capability to adapt pathology-specific knowledge effectively. Notably, PathGen-CLIP-L achieves impressive performance in the fully supervised setting, outperforming OpenAI-CLIP-L and approaching the results of GigaPath-G. This is remarkable given that GigaPath-G leverages much larger training datasets and model sizes.

Table 8: Performance comparison between zero-shot classification and fully supervised approaches.

| Model | WSSSLUAD | Lung | Colon | PatchCamelyon |
|---|---|---|---|---|
| Previous SOTA (Zero-shot) | 85.1 | 88.9 | 94.3 | 72.5 |
| PathGen-CLIP-L (Zero-shot) | 82.2 | 89.8 | 99.3 | 88.2 |
| OpenAI-CLIP-L (Full Fine-tuning) | 95.9 | 99.7 | 100.0 | 93.5 |
| GigaPath-G (Full Fine-tuning) | 97.1 | 100.0 | 100.0 | 96.9 |
| PathGen-CLIP-L (Full Fine-tuning) | 97.7 | 100.0 | 100.0 | 97.0 |

## B.5 Additional Ablations

### B.5.1 Ablations on the data curation process

We conduct an ablation study to evaluate the contribution of each step in our data curation process. As shown in the following table, removing either the clustering-based retrieval method or the prompt-based retrieval leads to a performance decline. The removal of clustering-based retrieval results in a more substantial average performance drop of 4.3%, likely because this method enhances dataset quality by sampling patches with diverse features. Eliminating the revise agent causes an average performance decrease of 2.3%, indicating its importance in correcting errors or hallucinations generated by large multimodal models. These findings underscore the critical role of each component in our data curation pipeline. Furthermore, by combining these approaches, PathGen-CLIP-L achieves an average performance improvement of 6.8% over PathGen-CLIP-L$_{init}$, demonstrating the effectiveness of the PathGen-1.6M dataset.

Table 9: Impact of each step in the PathGen-1.6M data curation process on the final performance of PathGen-CLIP, where RA represents the Revise Agent, PBR denotes prompt-based retrieval and CBR represents the clustering-based retrieval.

| Model | LC-Lung | LC-Colon | CRC100K | SkinCancer | Pcam | BACH | Osteo | WSSSLUAD | SICAPv2 | Average |
|---|---|---|---|---|---|---|---|---|---|---|
| PathGen-CLIP-L$_{init}$ | 89.1 | 96.9 | 60.3 | 55.7 | 83.2 | 65.3 | 71.9 | 85.1 | 48.3 | 72.9 |
| PathGen-CLIP-L w/o RA | 91.2 | 98.9 | 76.2 | 67.2 | 87.3 | 68.8 | 70.8 | 81.5 | 54.9 | 77.4 |
| PathGen-CLIP-L w/o PBR | 91.0 | 99.1 | 77.0 | 68.0 | 86.8 | 65.0 | 72.8 | 83.2 | 59.5 | 78.0 |
| PathGen-CLIP-L w/o CBR | 90.5 | 97.5 | 72.7 | 64.9 | 84.6 | 61.3 | 71.2 | 82.7 | 53.5 | 75.4 |
| PathGen-CLIP-L | 89.8 | 99.3 | 78.0 | 70.6 | 88.2 | 71.5 | 74.6 | 82.2 | 63.5 | 79.7 |

### B.5.2 Ablations on applying PathGen-1.6M to different CLIP models

To further demonstrate the broad applicability of PathGen-1.6M, we adapt it to other CLIP-like models, including the ConvNext-base-w320 (Liu et al., 2022), a convolution-based CLIP model trained by LAION (Schuhmann et al., 2022b), and a CLIPA-L (Li et al., 2023c), a more efficient architecture for training ViT-based CLIP models. As shown in Table 10, incorporating the PathGen-1.6M dataset significantly enhances the performance of both models, with improvements of 6.8% and 7.1%, respectively. These results further validate the quality and effectiveness of the PathGen-1.6M dataset.

### B.5.3 Ablations on the source of PathGen-1.6M's effectiveness

To explore whether the effectiveness of PathGen-1.6M arises from its high-quality image-text pairs, the images themselves, or the WSI reports used during its creation, we conduct two complementary

Table 10: Performance improvement of different CLIP models after continued training with PathGen-1.6M. Here, Model$_{init}$ represents models trained without PathGen-1.6M, while other models incorporate the PathGen-1.6M dataset during training.

| Model | LC-Lung | LC-Colon | CRC100K | SkinCancer | Pcam | BACH | Osteo | WSSSLUAD | SICAPv2 | Average |
|---|---|---|---|---|---|---|---|---|---|---|
| PathGen-CLIPA-L$_{init}$ | 79.3 | 98.4 | 61.5 | 50.5 | 86.1 | 56.5 | 59.8 | 78.5 | 48.0 | 68.7 |
| PathGen-CLIPA-L | 94.3 | 99.4 | 68.4 | 64.9 | 88.8 | 62.0 | 63.7 | 80.9 | 57.0 | 75.5 (+6.8) |
| PathGen-ConvNext-base$_{init}$ | 82.8 | 94.6 | 59.9 | 48.0 | 74.9 | 53.0 | 62.2 | 84.1 | 56.1 | 68.4 |
| PathGen-ConvNext-base | 89.1 | 95.9 | 67.9 | 60.2 | 80.7 | 66.2 | 77.9 | 87.2 | 54.3 | 75.5 (+7.1) |
| PathGen-CLIP-L$_{init}$ | 89.1 | 96.9 | 60.3 | 55.7 | 83.2 | 65.3 | 71.9 | 85.1 | 48.3 | 72.9 |
| PathGen-CLIP-L | 89.8 | 99.3 | 78.0 | 70.6 | 88.2 | 71.5 | 74.6 | 82.2 | 63.5 | 79.7 (+6.8) |

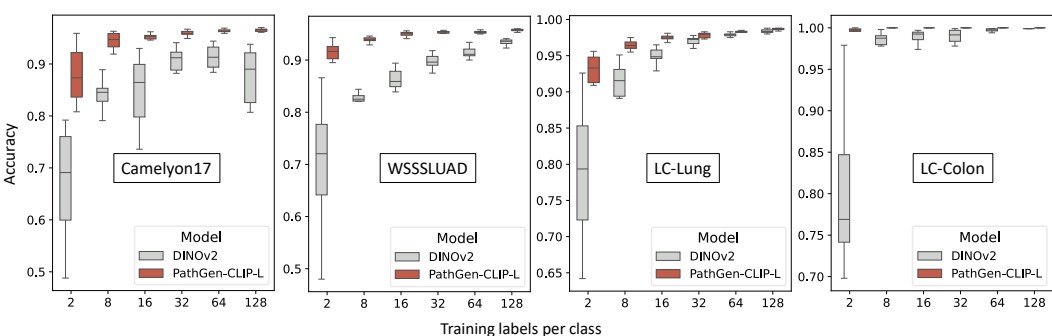

Figure 11: Few-shot classification performance comparison between PathGen-CLIP-L trained with PathGen-1.6M and DINO-V2-L trained using only the images from PathGen-1.6M.

experiments. In one approach, we replace all image captions in PathGen-1.6M with their corresponding WSI reports, creating "image-report" pairs to train PathGen-CLIP-L. Meanwhile, we also further fine-tune the pre-trained DINO-v2 (Oquab et al., 2023) by continuing pre-training on the images from PathGen-1.6M for 20 epochs as a comparison.

As shown in Table 11, the model trained with "image-report" pairs performs worse than PathGen-CLIP-L$_{init}$. This occurs because WSI reports provide a global summary that fails to accurately capture the localized features of individual patches. The mismatch between patch-level features and the shared global captions introduces significant challenges for CLIP's contrastive learning, as the model struggles to establish meaningful connections between image and text embeddings. In addition, Figure 11 reveals that DINO-v2 pretrained on PathGen-1.6M images significantly underperforms PathGen-CLIP-L in few-shot experiments. This is likely because DINO-v2 requires substantially larger datasets—on the scale of tens or even hundreds of millions of images—to achieve optimal performance. The 1.6 million images in PathGen-1.6M are insufficient for this method.

These experiments demonstrate that the superior performance of PathGen-CLIP-L primarily arises from the high-quality image-text pairs in PathGen-1.6M, underscoring the dataset's effectiveness.

Table 11: Zero-shot classification performance comparison by replacing PathGen-1.6M original captions with WSI reports.

| Model | LC-Lung | LC-Colon | CRC100K | SkinCancer | Pcam | BACH | Osteo | WSSSLUAD | SICAPv2 | Average |
|---|---|---|---|---|---|---|---|---|---|---|
| PathGen-CLIP-L (Image-report pairs) | 88.9 | 97.2 | 57.2 | 58.6 | 70.1 | 48.8 | 63.4 | 82.0 | 56.3 | 69.1 |
| PathGen-CLIP-L$_{init}$ | 89.1 | 96.9 | 60.3 | 55.7 | 83.2 | 65.3 | 71.9 | 85.1 | 48.3 | 72.9 |
| PathGen-CLIP-L | 89.8 | 99.3 | 78.0 | 70.6 | 88.2 | 71.5 | 74.6 | 82.2 | 63.5 | 79.7 |

## C EXPERIMENTAL DETAILS

### C.1 MORE DETAILS OF DATA CONSTRUCTION PROCESS

#### C.1.1 PATCH EXTRACTION DETAILS

When extracting image patches from the WSI of the TCGA, since many WSIs contain multiple layers with different magnifications, we always opt for the highest magnification layer to ensure the finest details and resolution are captured. The patches are consistently cropped to a size of $672 \times 672$ pixels.

#### C.1.2 DETAILS OF THE TRAINING DATA CONSTRUCTION PROCESS FOR THE REVISION AGENT

As we initially lack multimodal training samples for description revision (from incorrect to correct), we need to first construct such samples. As illustrated in Figure 12, we begin with a caption sentence presumed to be correct. We then employ a prompt with an LLM to generate a revision operation that intentionally introduces errors into the caption, creating an incorrect version. Next, we reverse this revision operation: additions are converted to deletions, deletions become additions, and for modifications, the sentences before and after the change are swapped. This process not only yields an incorrect caption but also the reverse operation that corrects the caption back to its original form, along with the corresponding image. These elements constitute our triplet revise agent training data.

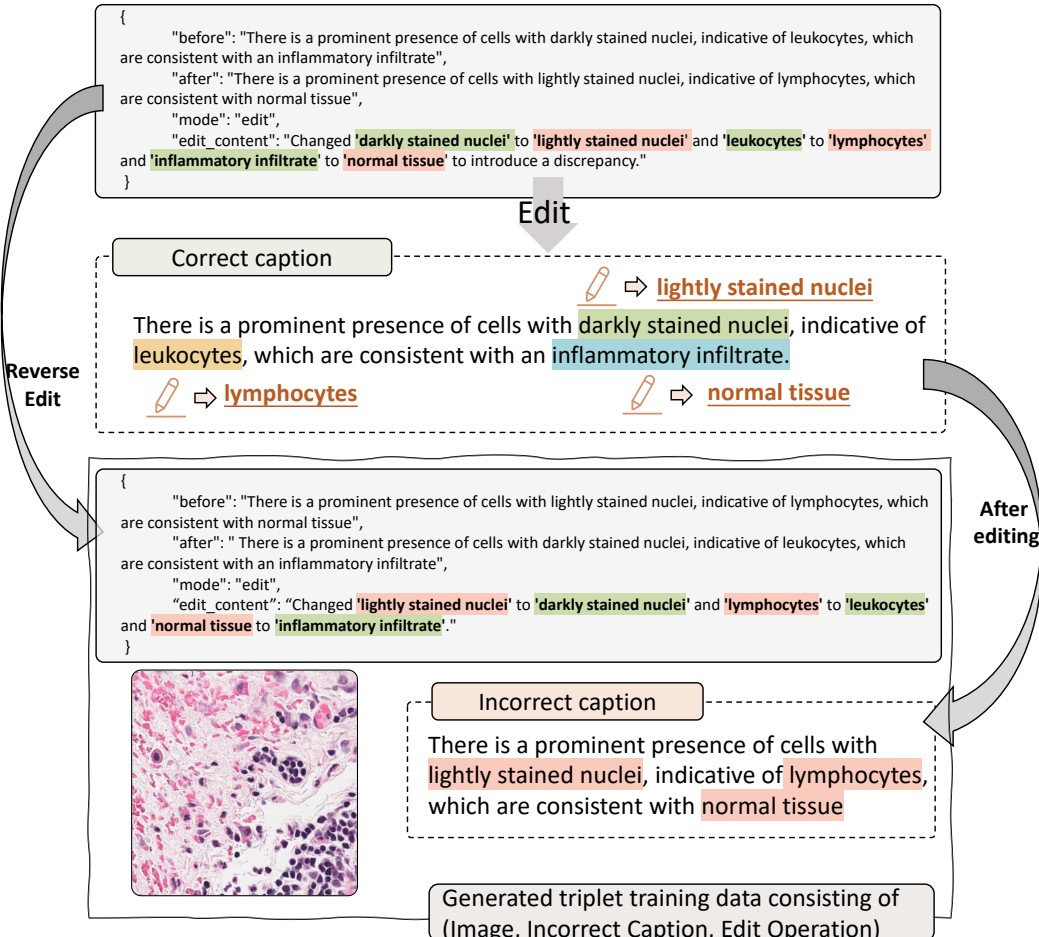

Figure 12: Illustration of the process for generating data on revision operations, using 'edit' as an example.

## C.2 PROMPTS

This section presents all the prompts used in our dataset and experimental process, including: (1) The prompt in Figure 13 is used with GPT-4V to add details and enhance existing image captions. (2) The prompt in Figure 14 is employed with GPT-4o to evaluate the quality of descriptions generated by two comparative multimodal models. (3) The prompt in Figure 15 is used to transform generated captions into erroneous versions and to specify the corresponding operations (add, delete, edit). By reversing this process, it generates a sequence of corrections that transition from erroneous to correct captions, serving as training data for the Revise Agent. For specific examples, please refer to Section C.1.2. (4) The prompt in Figure 16 generates open-ended QA instruction-tuning data based on existing descriptions. Additionally, Figure 17 is used to create multiple-choice QA instruction-tuning data. (5) The prompt in the left part of Figure 18 is used with GPT-4 to generate attribute-based prompts for PathGen-CLIP$_{init}$ (e.g., nuclear atypia, pleomorphism, stroma in lung WSI), and the right part of Figure 18 prompts GPT-4 to extract information related to pathological features from WSI reports, aiding in the generation of finding-based retrieval prompts for PathGen-CLIP$_{init}$. (6) We use the prompt "This is a histology image from the {source of tissue}. Describe this image in detail." to prompt PathGen-LLaVA$_{desp}$ in generating detailed descriptions of the images. In this prompt, "{source of tissue}" specifies the origin of the tissue sample.

## C.3 DOWNSTREAM TASKS AND DATASETS

### C.3.1 DATASETS FOR ZERO-SHOT AND FEW-SHOT IMAGE CLASSIFICATION

We employ nine datasets for tasks related to zero-shot and few-shot image classification, which include: **CRC100K** (Kather et al., 2018): This dataset comprises image patches taken from H&E stained histological samples and spans both colorectal cancer tissues and normal tissues. It is divided into nine distinct categories: Adipose, Background, Debris, Lymphocytes, Mucus, Smooth Muscle, Normal Colon Mucosa, Cancer-Associated Stroma, and Colorectal Adenocarcinoma Epithelium. **WSSS4LUAD** (Han et al., 2022): Featuring patch-level annotations from 87 whole slide images, this collection is focused on distinguishing between tumor and normal tissue classes. **LC25000** (Borkowski et al., 2019): This dataset includes samples of lung and colon adenocarcinomas, organized into two subsets: LC-lung, which encompasses lung adenocarcinomas, lung squamous cell carcinomas, and benign lung tissues; and LC-colon, which contains colon adenocarcinomas and benign colonic tissues. **PatchCamelyon** (Veeling et al., 2018): Originating from histopathological scans of lymph node sections, each image in this dataset carries a binary label indicating the presence or absence of metastatic tissue. **SICAPv2** (Silva-Rodríguez et al., 2020): This dataset features images of prostate pathology magnified 10 times, classified as non-cancerous, and Grades 3-5 according to the Gleason grading system. **BACH** (Aresta et al., 2019): Incorporating H&E stained breast histology images, this dataset categorizes images into four groups based on the dominant cancer type: normal, benign, in situ carcinoma, or invasive carcinoma. **Osteo** (Arunachalam et al., 2019): Sourced from whole slide images, this dataset aims to classify different tissue regions as viable tumors, necrotic tumors, or non-tumors, capturing the diverse responses of osteosarcoma to chemotherapy. **SkinCancer** (Kriegsmann et al., 2022): Consisting of tissue patches from skin biopsies across 12 anatomical compartments and 4 types of neoplasms, this dataset provides a comprehensive view of skin cancer variations.

### C.3.2 EXPERIMENT DETAILS OF LINEAR PROBING

The linear probe experiment is designed to evaluate the feature representation of a pre-trained model by adding a linear layer to its output. This linear layer maps the model's output vector to the number of classes for classification. We conduct the experiment using a batch size of 32 and run it for 20 epochs. The optimizer used is AdamW with a learning rate of $1 \times 10^{-2}$. To ensure robustness and reproducibility, we utilize 10 different seeds. The procedure involves randomly selecting 256 samples from each class to form the training set. If an official test set is unavailable or lacks labels, the remainder of the dataset serves as the test set. Throughout the 20 epochs, we select the best-performing model based on its accuracy on the test set.

### C.3.3 Datasets for WSI classification

This paper provides three commonly-used WSI classification datasets, CAMELYON16 (Litjens et al., 2018), CAMELYON17 (Litjens et al., 2018), and BRACS (Brancati et al., 2022), for evaluating PathGen-CLIP's representation quality for WSI classification, with rigorous and standardized splits to ensure robust model training and validation.

To be specific: **CAMELYON16** consists of 400 WSIs, with 270 assigned for training and 130 for testing. To enhance model validation, the training set is further divided into training and validation subsets in a 9:1 ratio, following the methodology in recent studies. **CAMELYON17** comprises 1,000 WSIs sourced from five different hospitals. These slides are categorized based on labels such as Normal, isolated tumor cells, Micro-metastases, and Macro-metastases. Due to the lack of labels in the official test set, the dataset's training set of 500 WSIs is reallocated to assess out-of-distribution (OOD) performance. Specifically, 200 WSIs from the fourth and fifth hospitals are designated as the test set, while the remaining 300 WSIs are split into training and validation sets in a 9:1 ratio. **BRACS (BReAst Carcinoma Subtyping)** includes 547 WSIs stained with Hematoxylin and Eosin (H&E), representing three lesion types: benign, malignant, and atypical. These are further divided into seven subcategories. Due to the limited number of WSIs, only three-class subtyping (benign, atypical, and malignant) is performed. The WSIs are segmented into non-overlapping patches of $224 \times 224$ at 20× magnification. The dataset is officially split into 395 training images, 65 validation images, and 87 test images, and this split is adhered to in subsequent analyses.

### C.3.4 Additional experimental details for WSI classification

**Data Pre-processing.** For the data pre-processing, we employ the method described in CLAM (Lu et al., 2021), which involves threshold segmentation and filtering to identify tissue regions within each whole-slide image (WSI). From these identified regions, we extract non-overlapping patches at a magnification of $\times 20$. Specifically, the crop size for the ViT-Base model is $224 \times 224$, whereas for the ViT-Large model, the crop size is $336 \times 336$.

**Feature Extraction.** This paper selects OpenAI CLIP (Radford et al., 2021), several of its variants specifically developed for the biomedical domain, and our PathGen-CLIP as a feature extractor. Features are extracted and saved with a dimension of 512 for ViT-Base and 768 for ViT-Large. In line with the regular practice of MIL methods, the feature extractor is frozen during training to save computational resources and storage.

**Model Architecture.** The MIL framework commonly used for WSI classification includes three learnable components: (1) A fully-connected layer to reduce the dimensionality of features to 256 for the ViT-Base model and 384 for the ViT-Large model. (2) An attention network to aggregate and transform the instance features. (3) A final fully-connected layer for making predictions. ABMIL (Ilse et al., 2018) and ACMIL (Zhang et al., 2023b) share the same fully-connected layers for reducing feature dimensionality and making predictions. For the attention network, ABMIL uses the gated attention network, and ACMIL introduces Multiple Branch Attention (MBA) and Stochastic Top-K Instance Masking (STKIM) based on the gated attention network.

**Training.** The models are trained for 50 epochs using a cosine learning rate decay schedule. The initial learning rates are determined through a grid search within the range $[0.0001, 0.0002, 0.0005]$, based on validation performance. The training process utilizes the Adam optimizer with a weight decay of 0.0001, and the batch size is consistently set to 1. Otherwise, we set $M = 5$, $K = 10$, and $p = 0.6$ for ACMIL.

### C.4 Hardware

We utilize 24 NVIDIA A100-80G GPUs for caption generation, 8 NVIDIA A100-80 G GPUs for training the PathGen-LLaVA model, 4 NVIDIA A100-80G GPUs for fine-tuning LLaMA, and 4 NVIDIA A100-40 G GPUs for training and testing on downstream datasets.

## C.5 TRAINING HYPERPARAMETERS

For the CLIP training, we adhere to the open_clip framework[2] and use OpenAI CLIP as initialization. We use a learning rate of 3e-5 with an Adam optimizer that includes a weight decay of 0.1. We set a batch size of 96 across 4 NVIDIA A100 GPUs, resulting in an effective batch size of 384. In the first stage of training using PathGen-1.6M, we limit the training to only one epoch. For the second stage of training with PathGen$_{init}$, we conduct two epochs.

For the training of PathGen-LLaVA, we use our trained PathGen-CLIP-L as the vision encoder and LLaVA-v1.5-13B (Liu et al., 2024) as the LLM component. We fully adhere to the training framework and parameters provided in LLaVA framework [3]. The training follows a two-stage process: in the first stage, we align the LLM with PathGen-CLIP-L using the PathGen$_{init}$ dataset, and in the second stage, we train using the PathGen-Instruct-200K dataset. We employ 8 NVIDIA A100-80G cards for the training of PathGen-LLaVA.

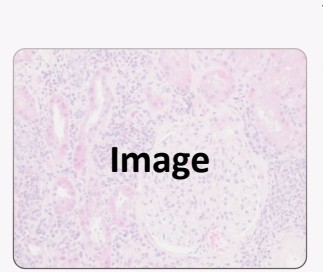

This is a microscopic image of cells or tissues. Please first describe the morphological characteristics of the cells or tissues in the image, and then supplement and correct it according to the additional description provided, but only include features that are observable in the image.

Be aware that some details in the additional description might not be present or identifiable in the image. Focus solely on the characteristics of the cells or tissues. Do not mention you have additional description such as 'considering the additional description', 'based on the given description'. Do not ask for other information or give unrelated response.

Additional description : {description}

Figure 13: Prompt for GPT-4V to add details and enhance existing image captions.

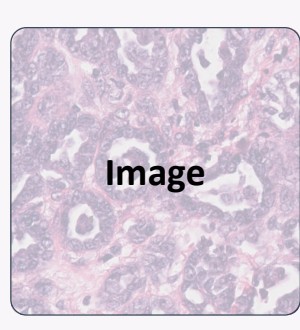

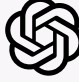

You need to evaluate the performance of two models in generating descriptions for a pathological image. The task for both models is to meticulously describe the visual pathological features evident in the provided image. Below are the responses from the two models. You are required to:

1. Carefully examine the provided pathological image and compare the responses from both models. Please focus on the description of the visual pathological features in the image and avoid placing too much emphasis on purely textual pathological expressions that are unrelated to the image.
2. Provide a comparison of the responses based on the given image, and analyze which model performed better. Conclude by stating which model is superior in the last line using the format "Model X is better" or "Both models are equally good" if you believe both models performed equally well.

Model A: {Caption generated by A}

Model B: {Caption generated by B}

Figure 14: Prompt for GPT-4o to evaluate the quality of descriptions generated by two comparative multimodal models.

[2] https://github.com/mlfoundations/open_clip
[3] https://github.com/haotian-liu/LLaVA

You need to modify the description generated by a multimodal model to intentionally introduce discrepancies that are inconsistent with the original image content described. The changes should adhere to the following requirements:

1. Modifications must be logically coherent and consistent with general pathological knowledge, but they should contradict or omit details relative to the described image content.
2. You need to specify the changes you make. For additions, include the sentence preceding the new content. Format the modifications as a JSON array, with each change detailed as follows:
   - For edits: {'before': 'original text', 'after': 'modified text', 'mode': 'edit', 'edit_content': 'description of change'}
   - For additions: {'before': '', 'after': 'added text', 'mode': 'add', 'previous_sentence': 'text before addition'}
   - For deletions: {'before': 'original text', 'after': '', 'mode': 'delete'}
3. Ensure there are 3-4 additions and deletions. Present the modifications in a JSON format like:
  {"changes": [change1, change2, ....] }

Description: {description}

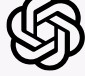

Figure 15: Prompt for GPT4 to transform generated captions into erroneous versions and to specify the corresponding operations (add, delete, edit).

This image is a pathological microscopic view of cells/tissues. Below is the image description.

Description: {description}

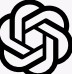

Please develop 2-5 high-quality, logically structured, challenging Q&A questions. Keep the following guidelines in mind:
1. Focusing on the cellular and tissue feature/morphology/diagnosis.
2. Frame the questions to resemble professional pathology exam questions, and ensure that each question has an answerable solution.
3. The questions should be designed that the answers are discernible only through careful observation of the image.
4. Ensure the answers are in detail and cannot be easily guessed.
5. Besides asking questions like "xxx feature suggests/indicative/consistent with  xxx," you must include questions that directly ask about the characteristics observed in the objects within the pathology image, such as "What features does the cell/tissue/xxx exhibit?"
6. Provide the questions along with their respective answers. Do not mention the answers are based on any provided description.\n\n'''+''' The answer should follow a JSON format: {"questions":[{"question": xxx, "answer": xxx}]}'''

Figure 16: Prompt for GPT4 to generate open-ended QA-based instruction-tuning data based on existing descriptions.

This image is a pathological microscopic view of cells/tissues. Below is the image description.

Description: {description}

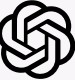

Please develop 1-3 high-quality multiple-choice questions. Keep the following guidelines in mind:
1. Focusing on the cellular and tissue feature/morphology/diagnosis.
2. Frame the questions to resemble professional pathology exam questions, and ensure that each question has an answerable solution.
3. Besides asking questions like "xxx feature suggests/indicative/consistent with  xxx," you must include questions that directly ask about the characteristics observed in the objects within the pathology image, such as "What features does the cell/tissue/xxx exhibit?"
4. The questions should be designed that the answers are discernible only through careful observation of the image.
5. The correct answer choices need to be deceptive; they cannot be easily guessed by just taking a shot in the dark.
6. Provide the questions along with their respective answers and explanations. Do not mention that the answers are based on any provided description.

The answer should follow a JSON format:
[{"question": xxx, "options": ['A) xxx', ....], "answer": xxx, "explanation": xxx}]'''

Figure 17: Prompt for GPT4 to generate multi-choice QA-based instruction-tuning data based on existing descriptions.

This image is a tissue patch from a {source of tissue} whole slide image of TCGA-{source of TCGA}. Please list 20 pathological features or attributes that might appear in this type of whole slide image.

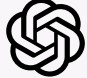

Your task is to extract descriptions of histomorphological characteristics evident in a whole slide image from the pathology report. The summary should:

1. Concentrate solely on the features visible in the microscopic image, disregarding any additional, non-visible details like descriptions of the gross morphology of the pathological specimen, clinical data, and measurements in centimeters (cm) and grams (g).
2. Retain the original terminology with high fidelity.
3. Refrain from using terms like 'mentioned,' 'description,' or 'report.'
4. Should the summary surpass 50 words, it must be divided into multiple segments, each not exceeding 50 words.
5. The summary must be presented in JSON format, structured as {"summary_part1": "<content>", "summary_part2": "<content>"}

Report: {WSI report}

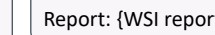

Figure 18: Prompt for generating attribute-based and finding-based text prompts for PathGen-CLIP, used to retrieve the most representative patches from WSIs.

Table 12: Classes for each dataset on zero-shot image classification. Note that we used the same prompt templates for each dataset. The templates used are: [ 'An H&E image of {}', 'this is an image of {} presented in image', 'An H&E patch of {}', ]

| Dataset | Classes |
|---|---|
| PatchCamelyon | 'lymph node', 'lymph node metastasis' |
| NCK-CRC | 'Adipose', 'Debris', 'Lymphocytes', 'Mucus', 'Smooth muscle', 'Normal colon mucosa', 'Cancer-associated stroma', 'Colorectal adenocarcinoma epithelium' |
| LC25000Lung | 'Lung adenocarcinoma', 'benign lung tissue', 'lung squamous cell carcinomas' |
| LC25000Colon | 'Colon adenocarcinoma', 'normal colon tissue' |
| BACH | 'Benign tissue', 'In-situ carcinoma', 'Invasive carcinoma', 'Normal tissue' |
| SICAPv2 | 'Non-cancerous', 'Atrophic well differentiated and dense glandular regions', 'Cribriform, ill-formed, large-fused and papillary glandular patterns', 'Isolated cells or file of cells, nests of cells without lumina formation and pseudo-rosetting patterns' |
| Osteo | 'Non-tumor', 'Necrotic tumor', 'Viable tumor' |
| SkinCancer | 'Non-tumor chondral tissue', 'Non-tumor dermis', 'Non-tumor elastosis', 'Non-tumor epidermis', 'Non-tumor hair follicle', 'Non-tumor skeletal muscle', 'Non-tumor necrosis', 'Non-tumor nerves', 'Non-tumor sebaceous glands','Non-tumor subcutis', 'Non-tumor sweat glands', 'Non-tumor vessel','Tumor epithelial basal cell carcinoma', 'Tumor epithelial squamous cell carcinoma', 'Tumor melanoma', 'Tumor naevus' |
| WSSS | 'tumor', 'normal' |

Table 13: Datasets used in our study and their corresponding source links

| Dataset | Source Link |
|---|---|
| PatchCamelyon17 | `https://patchcamelyon.grand-challenge.org/Download/` |
| CRC-100K | `https://zenodo.org/records/1214456` |
| SICAPv2 | `https://data.mendeley.com/datasets/9xxm58dvs3/1` |
| BACH | `https://iciar2018-challenge.grand-challenge.org/Dataset/` |
| Osteo | `https://journals.plos.org/plosone/article?id=10.1371/journal.pone.0210706` |
| SkinCancer | `https://heidata.uni-heidelberg.de/dataset.xhtml?persistentId=doi:10.11588/data/7QCR8S` |
| MHIST | `https://bmirds.github.io/MHIST` |
| WSSS4LUAD | `https://wsss4luad.grand-challenge.org/` |
| LC25000 (LC-Lung and LC-Colon) | `https://github.com/tampapath/lung_colon_image_set?tab=readme-ov-file` |
| BRCAS | `https://www.bracs.icar.cnr.it/` |
| Camelyon17 | `https://camelyon17.grand-challenge.org/Data/` |
| Camelyon16 | `https://camelyon16.grand-challenge.org/Data/` |
| PathMMU | `https://pathmmu-benchmark.github.io/#/` |

# D EXAMPLES

## D.1 GENERATED IMAGE-CAPTION PAIRS

Figures 20 to 24 showcase the descriptions we generated and the captions after summarization.

## D.2 GPT-4O EVALUATION EXAMPLES

Figures 25 to 28 showcase the evaluation of description quality generated by PathGen-LLaVA, Quilt-LLaVA, and LLaVA-Med-v1.5 using GPT-4o. These figures intricately demonstrate the specific strengths of our generated descriptions.

## D.3 PATHGEN-INSTRUCT EXAMPLES

Figures 29 and 30 showcase the multi-choice QA-based and open-ended-based instruction-tuning samples in PathGen-Instruct-200K.

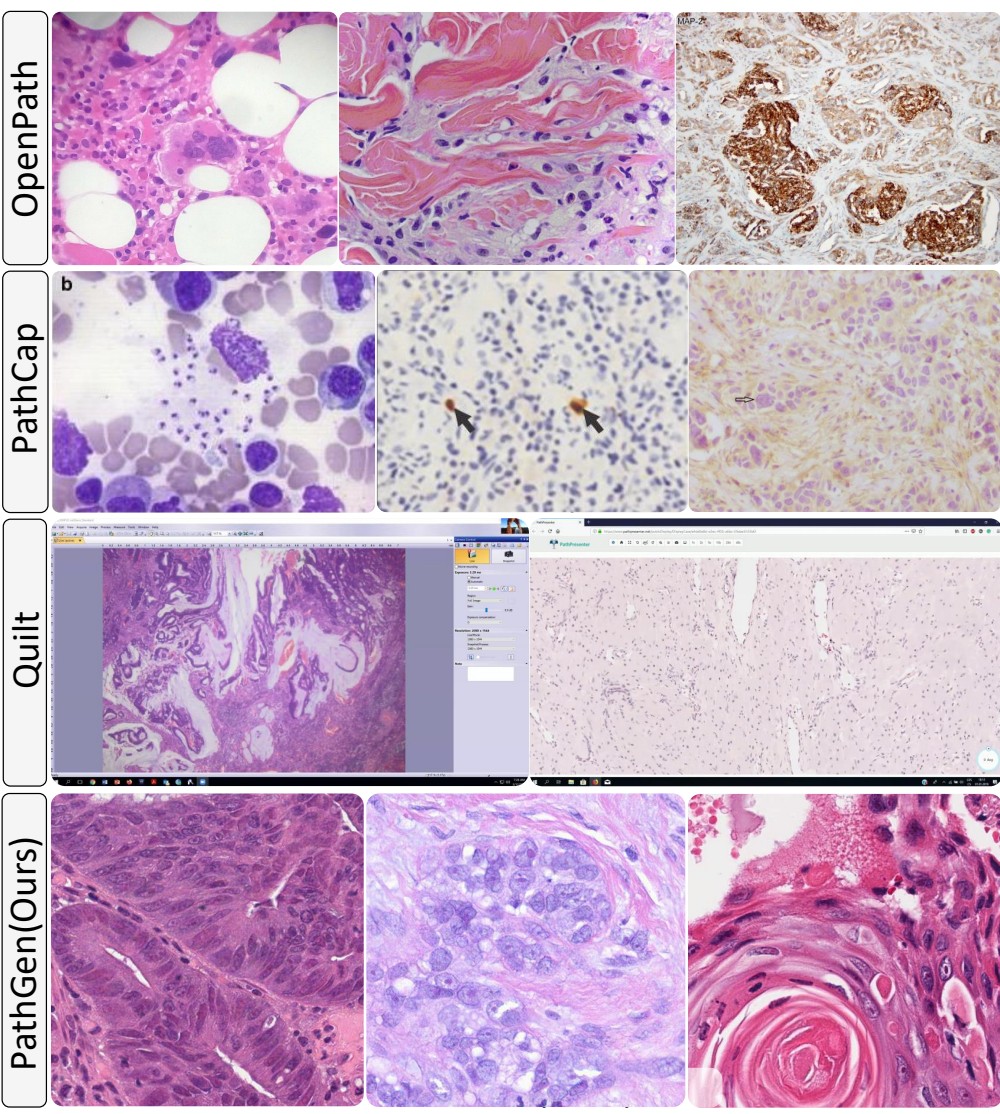

Figure 19: Comparison of image quality from different datasets: OpenPath, PathCap, Quilt, and PathGen.

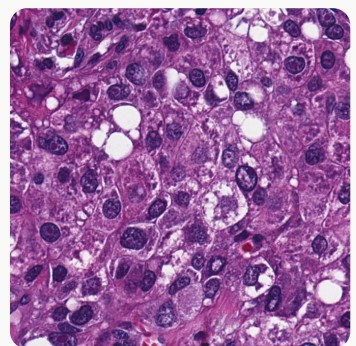

**Tissue origin: Liver**

**Summarized caption:** The liver tissue section displays hepatocytes with large, round nuclei and eosinophilic cytoplasm, some binucleated. Notable are vacuoles or lipid droplets in the cytoplasm. Sinusoids contain red blood cells; the architecture is largely intact with no signs of fibrosis, inflammation, or neoplastic changes. Further testing is required for a definitive diagnosis.

**Generated description:** The image shows a section of liver tissue with numerous cells exhibiting various morphological characteristics. The cells have large, round to oval nuclei with prominent nucleoli, which is typical for hepatocytes. Some of the cells contain multiple nuclei, suggesting they may be binucleated or multinucleated, which can be a normal variation in liver tissue. The cytoplasm of the cells appears eosinophilic, which is pinkish in color due to the staining technique used.

There are clear spaces within the cytoplasm of some cells, which could represent vacuolation or lipid droplets, a common feature in certain liver pathologies. The overall architecture of the tissue appears to be preserved, with no immediate signs of fibrosis or significant disruption of the normal lobular pattern.

The presence of red blood cells can be observed within the sinusoidal spaces, which are the small blood vessels within the liver. There is no obvious evidence of inflammatory infiltrate or neoplastic changes in this particular field of view. However, without additional context or staining, it is difficult to provide a definitive diagnosis or to identify subtle changes that might be present.

Figure 20: An example of a generated description and summarized caption for a given pathology image. Different pathological descriptions from various aspects are marked in distinct background colors.

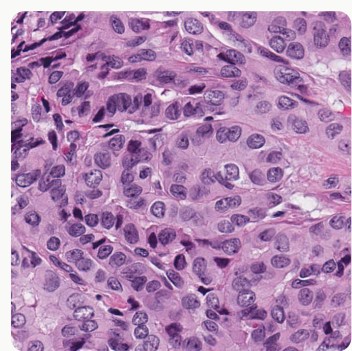

**Tissue origin: Thyroid**

**Summarized caption:** The histological section shows pink-staining collagenous stroma and cells with dark nuclei, likely lymphocytes, indicating inflammation. Larger cells suggest histiocytes. Absence of thyroid follicles suggests atrophy. These features, combined with the clinical history, may indicate Hashimoto's thyroiditis in a patient with Graves' disease and a diffuse goiter. Confirmation requires further clinical and immunohistochemical evaluation.

**Generated description:** The image shows a section of tissue with cells that have a somewhat irregular architecture. The cells appear to be follicular in nature, which is typical for thyroid tissue. They are organized in a pattern that is somewhat disrupted, with variable follicle sizes and shapes. The nuclei of the cells are enlarged and hyperchromatic, meaning they are darker than usual, suggesting a high nuclear-to-cytoplasmic ratio. There are also some nuclei that appear to have irregular contours and some with prominent nucleoli. The stroma, or connective tissue, between the follicles seems to be increased, which could indicate a fibrotic process. These features are suggestive of a neoplastic process, and given the clinical context of a thyroid nodule, could be consistent with a thyroid carcinoma. However, a definitive diagnosis would require clinical correlation and possibly additional immunohistochemical staining or molecular testing.

Figure 21: An example of a generated description and summarized caption for a given pathology image. Different pathological descriptions from various aspects are marked in distinct background colors.

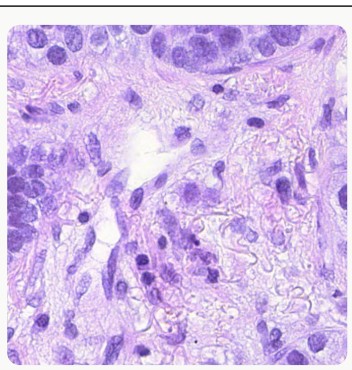

**Tissue origin: Lung**

**Summarized caption:** The lung tissue image displays cells with a high nucleus-to-cytoplasm ratio, hyperchromatic and pleomorphic nuclei, suggesting cellular atypia. Cells form glandular patterns, indicative of adenocarcinoma. The stroma is fibrous with inflammation. Differential diagnosis points to lung carcinoma, possibly adenocarcinoma, pending further clinical and molecular analysis for confirmation.

**Generated description:** The microscopic image shows a section of lung tissue with several morphological features. The tissue is composed of cells with a high nucleus-to-cytoplasm ratio, which is indicative of a high cellular density. The nuclei appear hyperchromatic and pleomorphic, suggesting variability in size and shape, which can be a sign of cellular atypia. There are also areas where the cells are arranged in a glandular or acinar pattern, which is typical of certain types of lung tissue, such as adenocarcinoma. The stroma, or connective tissue, appears to be fibrous with some inflammatory cells scattered throughout.

Based on the morphological characteristics observed in the image, a differential diagnosis could include a form of lung carcinoma, such as adenocarcinoma, given the glandular structures and cellular atypia. However, a definitive diagnosis would require clinical correlation and possibly additional immunohistochemical staining or molecular studies.

Figure 22: An example of a generated description and summarized caption for a given pathology image. Different pathological descriptions from various aspects are marked in distinct background colors.

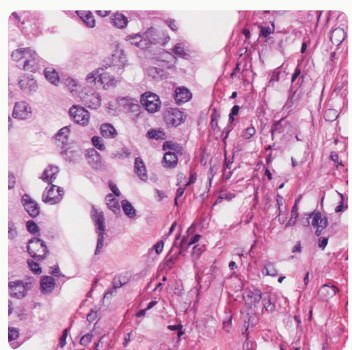

**Tissue origin: Bladder**

**Summarized caption:** The image shows malignant urothelial carcinoma cells with high nucleus-to-cytoplasm ratios, pleomorphic nuclei, and prominent nucleoli on the left. These cells infiltrate the organized, spindle-shaped cells of the muscularis propria on the right. The irregular cell shapes and disrupted tissue architecture indicate advanced disease, reflecting the cancer's penetration into deeper bladder layers.

**Generated description:** The image displays a section of tissue with various cellular and structural features. On the left side of the image, there is a cluster of cells with a high nucleus-to-cytoplasm ratio, pleomorphic nuclei, and prominent nucleoli, which are indicative of malignant cells. These cells are irregularly shaped and vary in size, suggesting a lack of normal cellular differentiation and tissue architecture.

The right side of the image shows a more organized structure with elongated cells that have a regular, spindle-shaped morphology. These cells are arranged in a parallel fashion, which is characteristic of smooth muscle or fibrous tissue. The boundary between the two areas is not sharply defined, indicating possible infiltration of the malignant cells into the adjacent tissue.

The overall features are consistent with a urothelial carcinoma, which is a common type of bladder cancer. The presence of atypical cells infiltrating into the muscularis propria suggests a higher stage of disease, as this indicates that the cancer has penetrated beyond the urothelial lining into the deeper muscle layer of the bladder wall.

Figure 23: An example of a generated description and summarized caption for a given pathology image. Different pathological descriptions from various aspects are marked in distinct background colors.

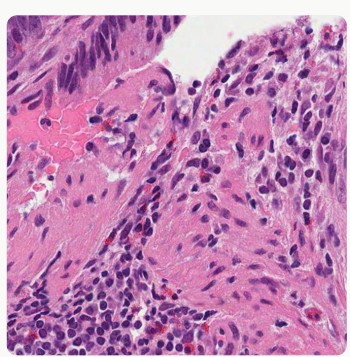

**Tissue origin: Colon**

**Summarized caption:** The colonic tissue image shows a mucosal layer with columnar epithelial cells and goblet cells, and a lamina propria with dense inflammatory infiltrate. Crypt architecture is distorted. These findings suggest chronic colitis. Differentiation between ulcerative colitis and Crohn's disease requires further clinical correlation and specific pathological characteristics like ulcers or granulomas.

**Generated description:** The image shows a section of colonic tissue with several notable features. The mucosal layer is visible with columnar epithelial cells lining the lumen. Goblet cells, which are mucus-secreting cells, can be seen interspersed between the epithelial cells. The lamina propria, which is the connective tissue layer beneath the epithelium, appears to have a dense infiltrate of inflammatory cells, including lymphocytes and plasma cells.

There is also evidence of crypt architectural distortion, with some crypts appearing irregular and branching, which is a sign of chronic inflammation. The submucosa appears to be expanded and contains more inflammatory cells, and there is a loss of the normal orderly arrangement of the crypts.

The presence of these features, particularly the crypt distortion and the dense inflammatory infiltrate in the lamina propria, is consistent with chronic colitis. The specific diagnosis of ulcerative colitis (UC) or Crohn's disease (CD) would typically require clinical correlation and additional features not visible in this image, such as ulcers or granulomas respectively.

Figure 24: An example of a generated description and summarized caption for a given pathology image. Different pathological descriptions from various aspects are marked in distinct background colors.

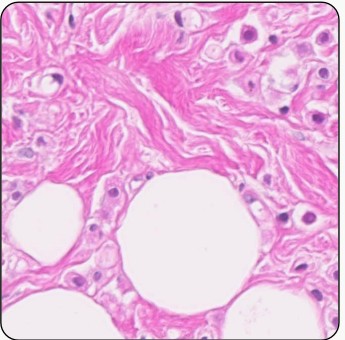

**Model A (Quilt-LLaVA):**
Upon examining the histopathology image of the breast, I observe a lobule that is not densely packed with cells. The cells within this lobule are predominantly epithelial cells, which are the primary constituents of the breast tissue.

In the upper left quadrant of the image, there is a significant presence of collagenous stroma. This stroma, which is the supportive framework of the breast tissue, appears to be interspersed with fibroblasts. These fibroblasts are cells that produce collagen and other fibers, contributing to the structural integrity of the tissue.

Moving towards the center and slightly to the right of the image, there is a region that is almost entirely composed of collagenous stroma. This area is characterized by a dense, fibrous structure, indicative of a high collagen content. The fibroblasts in this region appear to be less numerous, suggesting a mature and well-structured stromal component.

**Model B (PathGen-LLaVA):**
The microscopic image shows a section of breast tissue with several distinct features. There are multiple adipocytes, which are the large, white, empty-looking spaces that are characteristic of breast tissue. These cells have thin, peripheral nuclei due to the large vacuole of stored fat that pushes the nucleus to the side.

Interspersed between the adipocytes are fibrous stromal elements, which appear as pink, thread-like structures. These are collagen fibers providing structural support to the tissue.

Within the stroma, there are clusters of cells with enlarged, hyperchromatic nuclei and some degree of pleomorphism, which suggests an increased nuclear-to-cytoplasmic ratio. These cells are irregularly distributed and vary in size and shape, which is indicative of neoplastic changes.

The presence of these atypical cells within the breast tissue, along with the architectural disruption of the normal glandular structures, is suggestive of an invasive carcinoma. The cells' morphology, including their nuclear atypia and disorganized growth pattern, supports this diagnosis.

**GPT-4o:** **Comparison of Responses Based on the Given Image**
**Model A:**
**1. Description of Lobule and Epithelial Cells:**
1. Mentions a lobule that is not densely packed with cells and notes the presence of epithelial cells.
2. This description does not seem to directly correspond to the given image, where no clear lobules or concentrated epithelial cells are evident.
**2. Collagenous Stroma:**
1. Describes a significant presence of collagenous stroma in the upper left quadrant and centrally.
2. Correctly identifies the pink, fibrous structures as collagenous stroma and notes the presence of fibroblasts.
**Model B:**
**1. Adipocytes:**
1. Accurately identifies the large, white, empty-looking spaces as adipocytes, which characteristically contain vacuoles pushed to the side.
2. This is visible in the provided image, corresponding to the large spaces with thin peripheral nuclei.
**2. Fibrous Stromal Elements:**
1. Appropriately describes the pink, thread-like structures as fibrous stromal elements (collagen fibers).
**3. Cellular Features and Neoplastic Changes:**
1. Mentions clusters of cells with enlarged, hyperchromatic nuclei and some degree of pleomorphism, indicating nuclear atypia and neoplastic changes.
2. Suggests the presence of invasive carcinoma based on the description of these atypical cells.
3. The given image does show irregular nuclei, which aligns with this description.
**Analysis:**
•**Model A** provides a less accurate description of the image. The mention of epithelial cells and lobules does not reflect the actual contents depicted. Additionally, while it correctly identifies collagenous stroma, it adds details about fibroblasts which are not prominently seen or inferable from the image alone.
•**Model B** accurately describes the adipocytes and fibrous stroma, which are clearly visible in the image. It also identifies key pathological features such as enlarged, hyperchromatic nuclei and their disorganized pattern, indicative of neoplastic changes, possibly pointing towards invasive carcinoma. This description aligns closely with the visible histological features in the given image.
**Conclusion:**
Model B is better.

Figure 25: An example of GPT-4o's evaluation and comparison of generated descriptions by PathGen-LLaVA$_{desp}$ and Quilt-LLaVA.

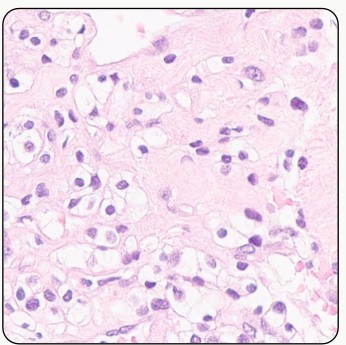

**Model A (PathGen-LLaVA):**
The microscopic image displays a section of renal tissue with several notable features. The cells exhibit a high degree of pleomorphism, with some cells having large, hyperchromatic nuclei and prominent nucleoli, which are indicative of malignant transformation. There is also evidence of abnormal mitotic figures, which are consistent with neoplastic activity. The architecture of the tissue is disrupted, lacking the organized structure typically seen in normal renal parenchyma. The presence of clear cells, which are large cells with abundant clear cytoplasm and distinct cell borders, is characteristic of clear cell renal cell carcinoma (RCC). The stroma appears to be sparse and vascularized, which is common in clear cell RCC. The overall features observed are consistent with a diagnosis of clear cell renal cell carcinoma.

**Model B (Quilt-LLaVA):**
The histopathology image reveals a kidney tissue sample with a notable presence of inflammatory cells. These cells are scattered throughout the tissue, indicating an ongoing inflammatory response. The cells are predominantly located in the interstitial spaces, which is a common site for inflammation in the kidney.

In addition to the interstitial inflammation, there is also evidence of inflammation within the tubules. This tubular inflammation is characterized by the infiltration of inflammatory cells into the tubular lumen. The tubules appear to be affected in a patchy manner, suggesting a non-uniform distribution of the inflammatory process.

The image also shows a few areas where the inflammation seems to be more pronounced. These areas are characterized by a higher density of inflammatory cells, indicating a more severe or active phase of inflammation. The overall picture is suggestive of a chronic inflammatory process, possibly due to an ongoing disease condition.

**GPT-4o:** Upon examining the provided pathological image, the following comparisons and analysis can be made about the responses from both models:
**Model A:**
•Describes the tissue as renal with specific emphasis on malignant features.
•Mentions high pleomorphism, hyperchromatic nuclei, and prominent nucleoli indicative of malignancy.
•Notes the presence of abnormal mitotic figures and disrupted architecture.
•Identifies clear cells with abundant clear cytoplasm, linking these features to clear cell renal cell carcinoma (RCC).
•Highlights sparse, vascularized stroma, characteristic of clear cell RCC, and concludes with a diagnosis of clear cell RCC.
**Model B:**
•Describes the tissue as kidney and mentions the presence of inflammatory cells.
•Focuses on the distribution of inflammatory cells in interstitial spaces and tubules.
•Describes the inflammation as patchy with some areas having a higher density of inflammatory cells.
•Concludes by suggesting a chronic inflammatory process possibly due to an ongoing disease condition.
**Comparison and Analysis:**
•**Accuracy in Describing Visual Features:**
  • **Model A**: Accurately identifies features such as pleomorphism, hyperchromatic nuclei, prominent nucleoli, abnormal mitotic figures, clear cells with abundant cytoplasm, and vascular stroma which are all visible in the image. It provides a cohesive description that matches the pathological features seen.
  • **Model B**: Mentions inflammatory cells and their distribution in interstitial spaces and tubules. While there is scattering of cells, the description does not match the image which heavily leans towards features of malignancy, specifically characteristics of clear cell RCC rather than chronic inflammation.
•**Overall Observation:**
  • **Model A** properly identified the specific type of pathology, matching its description with the visual details in the image such as clear cells and pleomorphism.
  • **Model B** provides a detailed account of inflammatory processes but does not align well with the image features that indicate malignant characteristics.
Based on the accuracy and relevance to the provided image:
**Model A is better**

Figure 26: An example of GPT-4o's evaluation and comparison of generated descriptions by PathGen-LLaVA$_{desp}$ and Quilt-LLaVA.

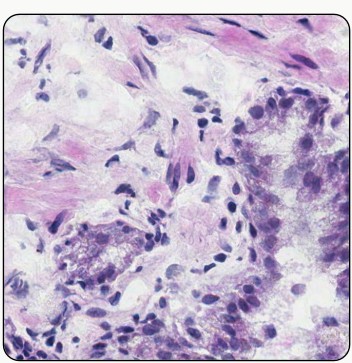

**Model A (Quilt-LLaVA):**
The histology image of the prostate reveals a number of intriguing features. The most striking aspect is the presence of nucleoli within the nuclei of the cells. These nucleoli are clearly visible, indicating a high level of cellular activity. However, it's important to note that the presence of nucleoli is not necessarily indicative of malignancy. In fact, it's quite common to observe nucleoli in prostate cells, even in a healthy state.

**Model B (PathGen-LLaVA):**
The image shows a section of prostate tissue with glandular structures that are irregularly shaped and lined by epithelial cells. The cells exhibit some variability in size and shape, with prominent nucleoli, which is suggestive of neoplastic changes. The stroma between the glands appears fibrous with a pinkish hue, indicating connective tissue. There is no clear evidence of invasion into the surrounding stroma by the glandular structures, which would be a feature suggestive of malignancy. However, the presence of architectural distortion and cellular atypia could be consistent with prostatic intraepithelial neoplasia (PIN), which is a precursor to prostate cancer.

**GPT-4o:** Upon examining the provided histology image of the prostate and comparing the responses, the following analysis can be made:
**Model A:**
• Describes the presence of nucleoli within the nuclei of the cells, which is accurate upon observation of the image.
• Highlights that nucleoli are indicative of a high level of cellular activity but not necessarily malignancy.
• Lacks detail about other features present in the image, such as glandular structures, stroma, or variability in cell size and shape.
**Model B:**
• Offers a more comprehensive description of the pathological features evident in the image.
• Accurately describes the presence of irregularly shaped glandular structures lined by epithelial cells and notes variability in cell size and shape.
• Comments on prominent nucleoli, suggestive of neoplastic changes, correlating well with the image.
• Includes observation of the fibrous stroma with a pinkish hue, indicating connective tissue presence.
• Mentions the lack of evidence of invasive malignancy yet observes architectural distortion and cellular atypia, suggesting possible prostatic intraepithelial neoplasia (PIN).
Given the image and the required detailed pathological description, Model B has provided a more thorough and accurate analysis of the visual features in the image. It covers more aspects of the tissue structure and offers insights into potential pathological conditions.
Model B is better.

Figure 27: An example of GPT-4o's evaluation and comparison of generated descriptions by PathGen-LLaVA$_{desp}$ and Quilt-LLaVA.

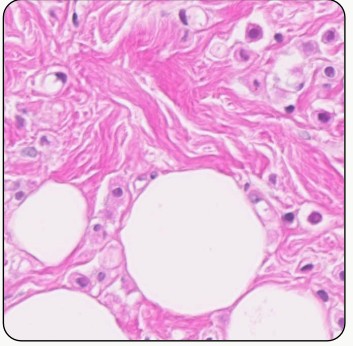

**Model A (Quilt-LLaVA):**
Upon examining the histopathology image of the breast, I observe a lobule that is not densely packed with cells. The cells within this lobule are predominantly epithelial cells, which are the primary constituents of the breast tissue.

In the upper left quadrant of the image, there is a significant presence of collagenous stroma. This stroma, which is the supportive framework of the breast tissue, appears to be interspersed with fibroblasts. These fibroblasts are cells that produce collagen and other fibers, contributing to the structural integrity of the tissue.

Moving towards the center and slightly to the right of the image, there is a region that is almost entirely composed of collagenous stroma. This area is characterized by a dense, fibrous structure, indicative of a high collagen content. The fibroblasts in this region appear to be less numerous, suggesting a mature and well-structured stromal component.

**Model B (PathGen-LLaVA):**
The microscopic image shows a section of breast tissue with several distinct features. There are multiple adipocytes, which are the large, white, empty-looking spaces that are characteristic of breast tissue. These cells have thin, peripheral nuclei due to the large vacuole of stored fat that pushes the nucleus to the side.

Interspersed between the adipocytes are fibrous stromal elements, which appear as pink, thread-like structures. These are collagen fibers providing structural support to the tissue.

Within the stroma, there are clusters of cells with enlarged, hyperchromatic nuclei and some degree of pleomorphism, which suggests an increased nuclear-to-cytoplasmic ratio. These cells are irregularly distributed and vary in size and shape, which is indicative of neoplastic changes.

The presence of these atypical cells within the breast tissue, along with the architectural disruption of the normal glandular structures, is suggestive of an invasive carcinoma. The cells' morphology, including their nuclear atypia and disorganized growth pattern, supports this diagnosis.

**GPT-4o:** **Comparison of Responses Based on the Given Image**
**Model A:**
**1.Description of Lobule and Epithelial Cells**:
  1. Mentions a lobule that is not densely packed with cells and notes the presence of epithelial cells.
  2. This description does not seem to directly correspond to the given image, where no clear lobules or concentrated epithelial cells are evident.
**2.Collagenous Stroma**:
  1. Describes a significant presence of collagenous stroma in the upper left quadrant and centrally.
  2. Correctly identifies the pink, fibrous structures as collagenous stroma and notes the presence of fibroblasts.
**Model B:**
**1.Adipocytes**:
  1. Accurately identifies the large, white, empty-looking spaces as adipocytes, which characteristically contain vacuoles pushed to the side.
  2. This is visible in the provided image, corresponding to the large spaces with thin peripheral nuclei.
**2.Fibrous Stromal Elements**:
  1. Appropriately describes the pink, thread-like structures as fibrous stromal elements (collagen fibers).
**3.Cellular Features and Neoplastic Changes**:
  1. Mentions clusters of cells with enlarged, hyperchromatic nuclei and some degree of pleomorphism, indicating nuclear atypia and neoplastic changes.
  2. Suggests the presence of invasive carcinoma based on the description of these atypical cells.
  3. The given image does show irregular nuclei, which aligns with this description.
**Analysis:**
•**Model A** provides a less accurate description of the image. The mention of epithelial cells and lobules does not reflect the actual contents depicted. Additionally, while it correctly identifies collagenous stroma, it adds details about fibroblasts which are not prominently seen or inferable from the image alone.
•**Model B** accurately describes the adipocytes and fibrous stroma, which are clearly visible in the image. It also identifies key pathological features such as enlarged, hyperchromatic nuclei and their disorganized pattern, indicative of neoplastic changes, possibly pointing towards invasive carcinoma. This description aligns closely with the visible histological features in the given image.
**Conclusion:**
Model B is better.

Figure 28: An example of GPT-4o's evaluation and comparison of generated descriptions by PathGen-LLaVA$_{desp}$ and LLaVA-Med-v1.5.

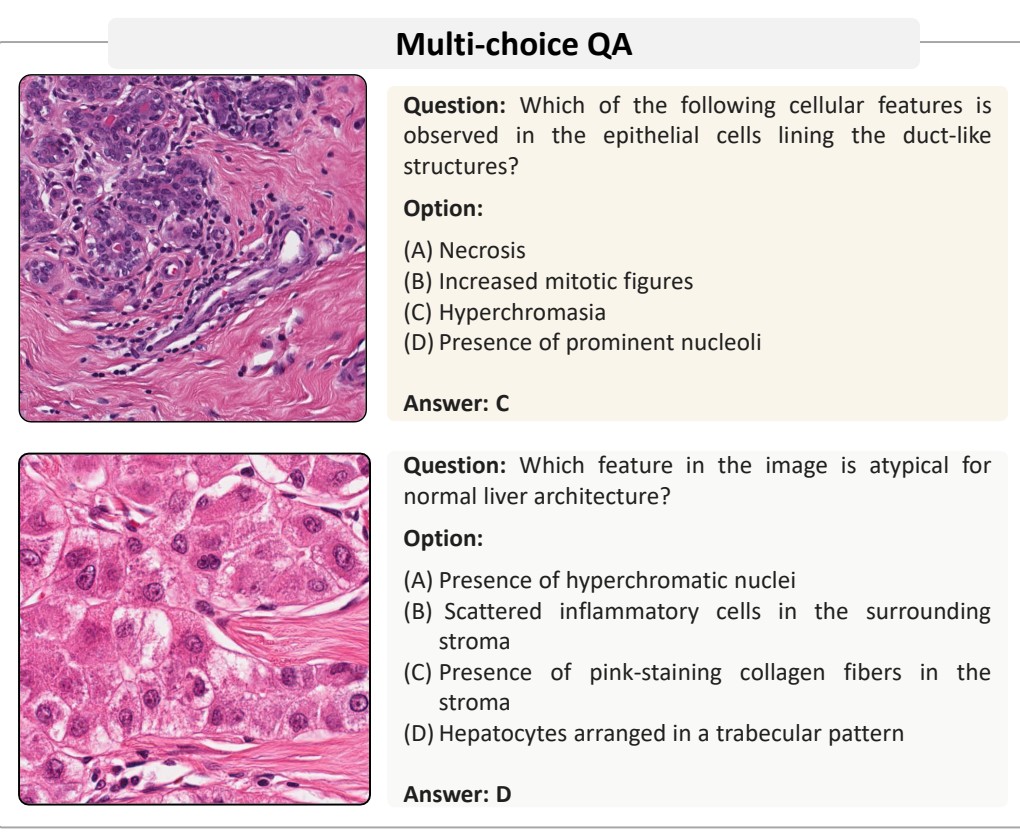

Figure 29: An example of a multi-choice QA-based instruction tuning sample from PathGen-Instruct-200K.

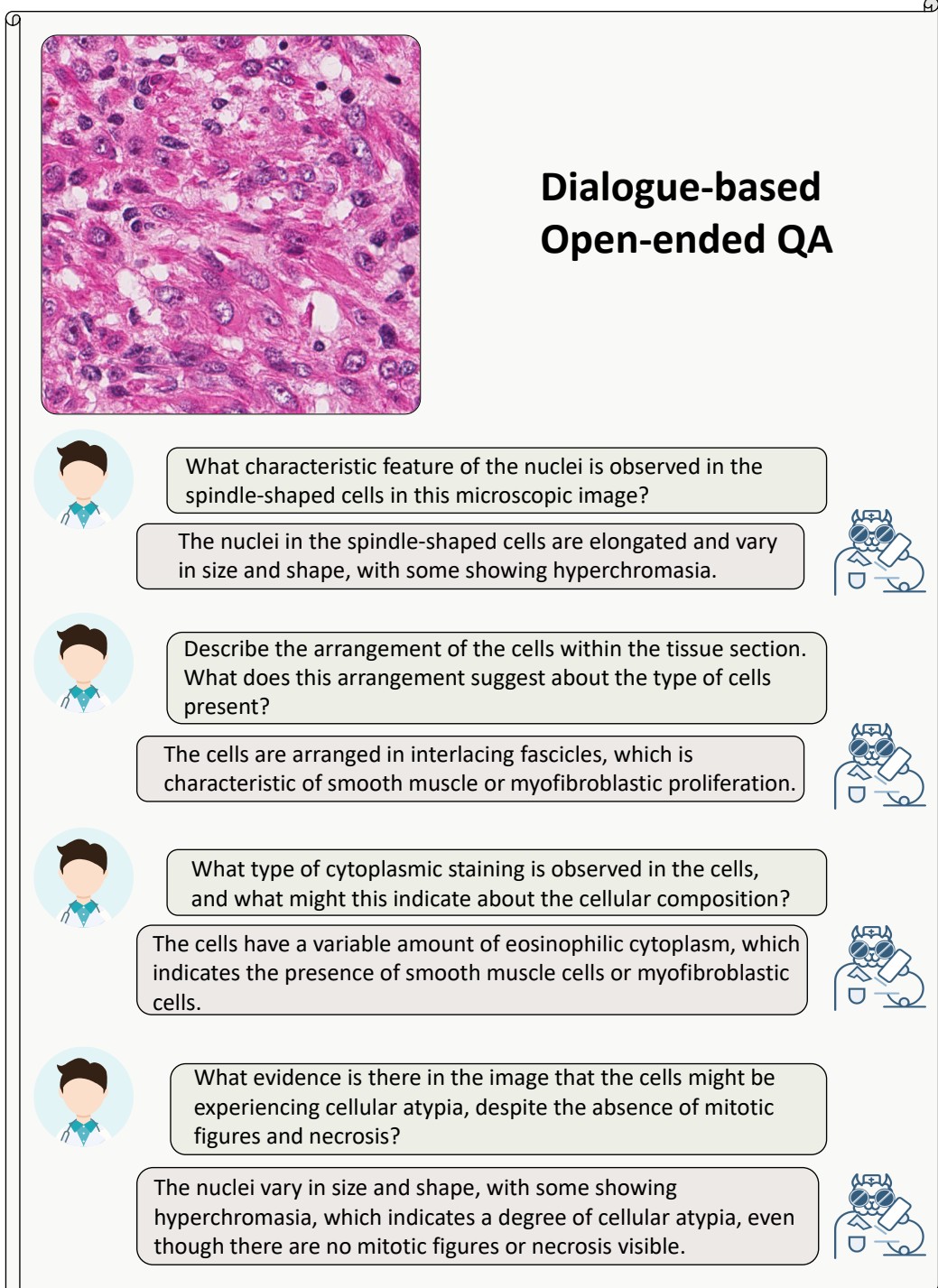

Figure 30: An example of a dialogue based open-ended QA instruction tuning sample from PathGen-Instruct-200K.

