# OpenReview forum: "PathGen-1.6M: 1.6 Million Pathology Image-text Pairs Generation through Multi-agent Collaboration"
_ICLR.cc/2025/Conference — ICLR 2025 Oral_

### Official Review · Reviewer_33XD · 2024-11-02

**Soundness:** 3
**Presentation:** 3
**Contribution:** 3
**Rating:** 6
**Confidence:** 5

**Summary:**

Built upon domain-specific WSI-Reports data from TCGA, this paper proposed the largest high-quality patch-text pairs dataset for Computational Pathology (CPath) by prompting LMM to generate text descriptions for pathology patches. To this end, a scalable data curation method is proposed by leveraging several LMM agents to describe, revise and summarize the generated descriptions.

**Strengths:**

1. To address data scarcity in CPath for pretraining large models, a scalable data curation method is proposed to expand the limited image-text dataset.
2. The largest patch-text pairs dataset is curated and used for pretraining to enhance the foundational power in CPath tasks.
3. The scalability to more WSI-only data is interesting, which has the potential to greatly expand the data scale for large pathology models.

**Weaknesses:**

1. To support the superiority of the proposed data construction method by introducing extra patch-level supervision, the performance of directly utilizing original report data along with patch images to pretrain a pathology foundation model should be presented.
2. During data construction, some representative patches were filtered out. These patches are supposed to align well with pathology reports, as they are retrieved based on the report data. To validate the extra contribution of generated patch descriptions, the proposed model should be compared with the one trained on these selected representative patches as well as its corresponding report data.
3. The model’s generalizability to out-of-domain data has not been validated. The authors tried to scale the model to non-WSI report paired data. However, these data are still from TCGA.
4. Some SOTA pathology foundation models are missing, especially in experimental comparison, such as UNI [1], CONCH[2], mSTAR[3] (which is also a CLIP-style model trained on TCGA data as well), etc.
5. Few-shot’s capability significantly relies on how well-aligned the vision and text spaces are. The proposed method is supposed to compare with VLM like CONCH, instead of vision-only GigaPath.
6. Details in EXPERT EVALUATION are missing. For example, how do authors define what correct or incorrect findings are?

Minor concerns:

- It is hard to recognize different models according to their colors. Please choose ones with higher contrast.
- The citation of GigaPath seems to be lost.

[1] Towards a general-purpose foundation model for computational pathology, Nature Medicine, 2024

[2] A visual-language foundation model for computational pathology, Nature Medicine, 2024

[3] A Multimodal Knowledge-enhanced Whole-slide Pathology Foundation Model, arxiv, 2024

**Questions:**

1. In Step 5 of data construction, how do authors ensure that no essential details are lost?
2. How can be validated the effectiveness of each step in data curation? This should be discussed in the ablation study.

---

> ### Author Response · Authors · 2024-11-24
> **Response to Reviewer 33XD (1/2)**
>
> We sincerely thank you for your valuable suggestions and recognition of our work. Please refer to our responses to your suggestions below.
>
> > ### The performance of directly utilizing original report data along with patch images to pretrain a pathology foundation model should be presented.
>
> Thanks for the suggestion. On one hand, WSI reports provide a global summary and cannot accurately represent the features of each local patch. This mismatch between patch-level features and shared captions introduces significant challenges for CLIP's contrastive learning, as the model struggles to establish meaningful associations between image and text embeddings.
>
> Moreover, the dataset contains only about 7,500 WSI reports, which further limits the model's ability to learn more diverse features. To illustrate this point, we trained **PathGen-CLIP-L** by using the WSI report as the caption for every patch in the WSI. As shown in the following table, the model trained in this manner performs even worse than **PathGen-CLIP-L_init**.
>
> | **Model**                           | **LC-Lung** | **LC-Colon** | **CRC100K** | **SkinCancer** | **Pcam** | **BACH** | **Osteo** | **WSSSLUAD** | **SICAPv2** | **Average** |
> | ----------------------------------- | ----------- | ------------ | ----------- | -------------- | -------- | -------- | --------- | ------------ | ----------- | ----------- |
> | PathGen-CLIP-L (Image-report paris) | 88.9        | 97.2         | 57.2        | 58.6           | 70.1     | 48.8     | 63.4      | 82.0         | 56.3        | 69.1        |
> | PathGen-CLIP-L_init                 | 89.1        | 96.9         | 60.3        | 55.7           | 83.2     | 65.3     | 71.9      | **85.1**     | 48.3        | 72.9        |
> | PathGen-CLIP-L                      | **89.8**    | **99.3**     | **78.0**    | **70.6**       | **88.2** | **71.5** | **74.6**  | 82.2         | **63.5**    | **79.7**    |
>
> > ### To validate the extra contribution of generated patch descriptions, the proposed model should be compared with the one trained on these selected representative patches as well as its corresponding report data.
>
> We sincerely appreciate your suggestion, and in response, we conducted additional DINO-v2 pretraining on the images from PathGen-1.6M. As shown in the following tables, DINO-v2 pretrained on PathGen-1.6M images significantly underperforms PathGen-CLIP-L in few-shot experiments. This is likely because DINO-v2 requires substantially larger samples—on the scale of tens or even hundreds of millions of images—to achieve optimal performance. In contrast, this highlights that the high-quality image-text pairs in PathGen-1.6M are key to enhancing the model's capabilities, serving as a more effective approach than vision-only self-supervised learning approach like DINO.
>
> | **Dataset: Colon** | **8**     | **16**    | **32**   | **64**    | **128**   | **256**   |
> | ------------------ | --------- | --------- | -------- | --------- | --------- | --------- |
> | DINOv2-L           | 98.2      | 99.0      | 99.0     | 99.7      | 99.9      | 99.9      |
> | PathGen-CLIP-L     | **100.0** | **100.0** | **99.9** | **100.0** | **100.0** | **100.0** |
>
> | **Dataset: LC-Lung** | **8**    | **16**   | **32**   | **64**   | **128**  | **256**  |
> | -------------------- | -------- | -------- | -------- | -------- | -------- | -------- |
> | DINOv2-L             | 90.7     | 95.0     | 97.1     | 97.9     | 98.4     | 99.0     |
> | PathGen-CLIP-L       | **96.5** | **97.5** | **97.8** | **98.1** | **98.7** | **99.1** |
>
> | **Dataset: WSSSLUAD** | **8**    | **16**   | **32**   | **64**   | **128**  | **256**  |
> | --------------------- | -------- | -------- | -------- | -------- | -------- | -------- |
> | DINOv2-L              | 82.5     | 86.4     | 89.8     | 91.4     | 93.4     | 94.6     |
> | PathGen-CLIP-L        | **93.9** | **95.0** | **95.2** | **95.3** | **95.7** | **95.7** |
>
> | **Dataset: PatchCamelyon** | **8**    | **16**   | **32**   | **64**   | **128**  | **256**  |
> | -------------------------- | -------- | -------- | -------- | -------- | -------- | -------- |
> | DINOv2-L                   | 83.7     | 85.0     | 90.4     | 90.8     | 87.7     | 91.3     |
> | PathGen-CLIP-L             | **94.5** | **94.4** | **95.9** | **96.4** | **96.6** | **96.6** |

---

> > ### Comment · Reviewer_33XD · 2024-11-26
> >
> > Thanks for your responses, which address the majority of my concerns.
> > However, it seems that the authors misunderstood the validation of the extra contribution of generated patch descriptions.
> > My concern is that a PathGen-CLIP-L trained on the selected representative patches and their paired reports should be compared, instead of vision-only DINOv2 trained on the selected representative patches.

---

> ### Author Response · Authors · 2024-11-24
> **Response to Reviewer 33XD (2/2)**
>
> > ### Some SOTA pathology foundation models are missing, especially in experimental comparison, such as UNI [1], CONCH[2], mSTAR[3] (which is also a CLIP-style model trained on TCGA data as well), etc.
>
> Thank you for your suggestion. As UNI is a vision-only model and mSTAR has not yet been open-sourced, we focused on comparing our method with more similar model, CONCH. This comparison includes zero-shot classification, few-shot classification, and WSI classification tasks. **PathGen-CLIP-L** consistently demonstrates better performance than CONCH across these tasks. For detailed results, please kindly refer to the General Response.
>
> > ### Few-shot’s capability significantly relies on how well-aligned the vision and text spaces are. The proposed method is supposed to compare with VLM like CONCH, instead of vision-only GigaPath.
>
> Thank you for your suggestion. We provide a detailed comparison of the few-shot capabilities between our proposed method and CONCH in the General Response. Please kindly refer to the General Response for further details.
>
> > ### Details in EXPERT EVALUATION are missing. For example, how do authors define what correct or incorrect findings are?
>
> Thank you for your comment. In our expert evaluation, findings were assessed in a straightforward manner based on two criteria: (1) whether the description aligns with observable image features (e.g., stating there is necrosis when only scattered mucin is visible would be incorrect); and (2) whether the description uses standard professional pathological terminology (rare cases where pathologists could not understand the findings were deemed incorrect).
>
> > ### In Step 5 of data construction, how do authors ensure that no essential details are lost?
>
> In step 5, the description agent's outputs tend to be verbose, and the summary agent focuses on condensing these overly detailed sentences into concise summaries without losing essential information. Additionally, during the prompt engineering for GPT-4, we actively incorporated feedback from pathologists, iteratively refining the prompts until they produced results that met their satisfaction. Occasionally, certain details, such as color or HE staining information, may be omitted during this process. However, these details are not critical for accurate diagnosis and do not impact the overall effectiveness of the findings.
>
> > ### How can be validated the effectiveness of each step in data curation? This should be discussed in the ablation study.
>
> Thank you for your insightful feedback. Following your suggestion, we conducted an ablation study to validate the effectiveness of each step in our data curation process. As illustrated in following table, removing either the clustering-based retrieval method or the prompt-based retrieval both led to a decline in performance.
>
> - Notably, the removal of the clustering-based retrieval resulted in a more substantial performance decrease of 4.3%. We hypothesize that this is because clustering-based retrieval is capable of sampling patches with more diverse features, thereby enhancing the overall dataset quality.
> - Additionally, eliminating the revise agent caused an average performance drop of 2.3%, indicating its effectiveness in correcting errors or hallucinations generated by LMMs.
> - These findings confirm that each component of our data curation pipeline is crucial.
>
> | **Model**                                    | **LC-Lung** | **LC-Colon** | **CRC100K** | **SkinCancer** | **Pcam** | **BACH** | **Osteo** | **WSSSLUAD** | **SICAPv2** | **Average** |
> | -------------------------------------------- | ----------- | ------------ | ----------- | -------------- | -------- | -------- | --------- | ------------ | ----------- | ----------- |
> | PathGen-CLIP-L wo/ Revise agent              | 91.2        | 98.9         | 76.2        | 67.2           | 87.3     | 68.8     | 70.8      | 81.5         | 54.9        | 77.4        |
> | PathGen-CLIP-L wo/ prompt-based retrieval    | 91.0        | 99.1         | 77.0        | 68.0           | 86.8     | 65.0     | 72.8      | 83.2         | 59.5        | 78.0        |
> | PathGen-CLIP-L wo/ clusering-based retrieval | 90.5        | 97.5         | 72.7        | 64.9           | 84.6     | 61.3     | 71.2      | 82.7         | 53.5        | 75.4        |
> | PathGen-CLIP-L                               | 89.8        | 99.3         | 78.0        | 70.6           | 88.2     | 71.5     | 74.6      | 82.2         | 63.5        | 79.7        |

---

> ### Author Response · Authors · 2024-11-26
> **Clarification on the Validation of Extra Contribution from Generated Patch Descriptions**
>
> **Dear Reviewer 33XD,**
>
> Thank you for taking the time to read our response. We are glad to hear that the majority of your concerns have been addressed.
>
> Regarding your concern about the validation of the extra contribution of generated patch descriptions: **In our first response,** we conducted experiments to demonstrate that **PathGen-CLIP-L trained on the selected representative patches and their ''paired reports''** leads to significant **performance degradation (-10.6%)** compared to **PathGen-CLIP-L trained on the generated captions** (we attach the table below for reference).
>
> This performance degradation occurs because the **patches themselves do not have corresponding reports. Instead, the WSI, which contains thousands of patches, is paired with a single report.** By attempting to pair individual patches with the WSI report, it will introduce **significant misalignment and a many-to-one relationship**, which adversely affects the contrastive learning in CLIP.
>
> For the subsequent DINO experiments, we conducted additional experiments to validate the effectiveness of PathGen-1.6M from a different perspective (image-only) to further demonstrate the improvement is primary from **PathGen-1.6M's image-caption pairs**.
>
> We hope this explanation addresses your concern, and we look forward to your reply. If we have misunderstood your question in any way, we would be more than happy to conduct further experiments to validate your concern.
>
> | **Model**                                          | **LC-Lung** | **LC-Colon** | **CRC100K** | **SkinCancer** | **Pcam** | **BACH** | **Osteo** | **WSSSLUAD** | **SICAPv2** | **Average** |
> | -------------------------------------------------- | ----------- | ------------ | ----------- | -------------- | -------- | -------- | --------- | ------------ | ----------- | ----------- |
> | PathGen-CLIP-L **(trained on patch-report paris)** | 88.9        | 97.2         | 57.2        | 58.6           | 70.1     | 48.8     | 63.4      | 82.0         | 56.3        | 69.1        |
> | PathGen-CLIP-L_init                                | 89.1        | 96.9         | 60.3        | 55.7           | 83.2     | 65.3     | 71.9      | **85.1**     | 48.3        | 72.9        |
> | PathGen-CLIP-L                                     | **89.8**    | **99.3**     | **78.0**    | **70.6**       | **88.2** | **71.5** | **74.6**  | 82.2         | **63.5**    | **79.7**    |
>
> Thank you once again!

---

### Official Review · Reviewer_Y3n6 · 2024-11-03

**Soundness:** 3
**Presentation:** 3
**Contribution:** 3
**Rating:** 8
**Confidence:** 5

**Summary:**

This paper introduces PathGen-1.6M, a large-scale dataset containing 1.6 million high-quality image-caption pairs extracted from Whole Slide Images (WSI).  It also developed a scalable approach for high-quality pathology image-text data generation by multiple agent models collaborating, paving the way for next-generation general pathology models.

**Strengths:**

1.	The paper presents PathGen-1.6M, a dataset containing 1.6 million high-quality image-caption pairs. Based on this dataset, the authors develop PathGen-CLIP, a pathology-specific CLIP model, which achieves substantial improvements across nine pathology-related zero-shot image classification tasks and three whole-slide image tasks.
2.	The authors propose a data construction pipeline that employs multiple LLM agents for description, revision, and summarization. This multi-agent collaboration approach generates more accurate image-caption pairs, validated through human evaluation.
3.	The experiments conducted in the paper are solid and well-executed.
4.	The release of the dataset, code, and model contributes significantly to the advancement of the pathology image research community.

**Weaknesses:**

1.	The Revise LMM appears to have limited utility, primarily providing editing capabilities such as additions, deletions, or modifications. It seems ineffective in addressing common pathological inaccuracies in the generated descriptions.
2.	Only evaluated in CLIP model. Can be work in more pretrained Vision-Language model.

**Questions:**

1.	The writing of the paper requires improvement, as there are several shortcomings. For example, there is a typo in “PathGen-LLaVAdesp” in Section 4.1, and the conclusion summarizes, “we train two advanced models: PathGen-CLIP and PathGen-CLIP-L.” It appears there is an additional model, “PathGen-LLaVA,” which needs clarification.
2.	A significant concern arises regarding potential data leakage in the downstream tasks and benchmarks evaluated. To my knowledge, many tasks and benchmarks are derived from TCIA pathology data, which raises suspicions about the homogeneity of the constructed dataset.

---

> ### Author Response · Authors · 2024-11-24
> **Response to Reviewer Y3n6 (1/2)**
>
> We deeply appreciate your thoughtful and constructive feedback. We have carefully addressed each of your concerns in detail and incorporated your suggestions to strengthen our manuscript.
>
> > ### It seems ineffective in addressing common pathological inaccuracies in the generated descriptions.
>
> Thank you for your thoughtful question. The Revise Agent is specifically designed to address certain inaccuracies or potential hallucinations in the generated descriptions. For instance, it can help correct errors such as mistaking scattered mucus for necrosis or overdiagnosing inflammatory regions as cancer. While it may not completely eliminate all inaccuracies, it does provide a significant improvement compared to making no corrections at all.
>
> To evaluate the effectiveness of the Revise Agent, we conducted a benchmark comparing the zero-shot performance of CLIP with and without the Revise Agent. The results indicate that the Revise Agent improves overall performance by 2.3%. This demonstrates its ability to enhance data quality by generating more accurate captions, which in turn strengthens the model's capabilities.
>
> | **Model**                       | **LC-Lung** | **LC-Colon** | **CRC100K** | **SkinCancer** | **Pcam** | **BACH** | **Osteo** | **WSSSLUAD** | **SICAPv2** | **Average** |
> | ------------------------------- | ----------- | ------------ | ----------- | -------------- | -------- | -------- | --------- | ------------ | ----------- | ----------- |
> | PathGen-CLIP-L w/o Revise agent | **91.2**    | 98.9         | 76.2        | 67.2           | 87.3     | 68.8     | 70.8      | 81.5         | 54.9        | 77.4        |
> | PathGen-CLIP-L                  | 89.8        | **99.3**     | **78.0**    | **70.6**       | **88.2** | **71.5** | **74.6**  | **82.2**     | **63.5**    | **79.7**    |
>
> > ### The writing of the paper requires improvement. For example, there is a typo in “PathGen-LLaVAdesp” in Section 4.1, and the conclusion summarizes, “we train two advanced models: PathGen-CLIP and PathGen-CLIP-L.” It appears there is an additional model, “PathGen-LLaVA,” which needs clarification.
>
> We apologize for any potential confusion caused, as there are many intermediate agents and the final trained model, which might have led to some misunderstanding. Specifically, PathGen-LLaVAdesp is not our final model but rather an intermediate model used exclusively for generating image descriptions.
>
> Overall, our final output models are of two types: a CLIP series (**PathGen-CLIP and PathGen-CLIP-L**, trained using the PathGen-1.6M image-caption dataset) and a large multimodal model (**PathGen-LLaVA**, trained using PathGen-instruct 200k, capable of handling instruction-following tasks such as visual question answering, classification, and captioning).
>
> The sentence we mentioned, "we train two advanced models: PathGen-CLIP and PathGen-CLIP-L," it specifically referred to the two different sizes of the CLIP versions.
>
> > ### A significant concern arises regarding potential data leakage in the downstream tasks and benchmarks evaluated. To my knowledge, many tasks and benchmarks are derived from TCIA pathology data, which raises suspicions about the homogeneity of the constructed dataset.
>
> Thank you for your valuable feedback. To address concerns about data leakage and ensure dataset diversity, we have taken deliberate measures to avoid overlap with TCGA/TCIA subtyping datasets, such as TCGA-BRCA and TCGA-RCC, in our downstream tasks. Instead, we selected datasets like **BRACS** (a different dataset from TCGA-BRCA), **Camelyon17**, **Camelyon16**, and **PathMMU**, none of which are derived from TCGA/TCIA.
>
> This careful selection minimizes homogeneity and ensures the robustness and generalizability of our results. Below, we provide the source links for each dataset for your reference:
>
> - PatchCamelyon17: https://patchcamelyon.grand-challenge.org/Download/
> - CRC-100K: https://zenodo.org/records/1214456
> - SICAPv2: https://data.mendeley.com/datasets/9xxm58dvs3/1
> - BACH: https://iciar2018-challenge.grand-challenge.org/Dataset/
> - Osteo: https://journals.plos.org/plosone/article?id=10.1371/journal.pone.0210706
> - SkinCancer: https://heidata.uni-heidelberg.de/dataset.xhtml?persistentId=doi:10.11588/data/7QCR8S
> - MHIST: https://bmirds.github.io/MHIST
> - WSSS4LUAD: https://wsss4luad.grand-challenge.org/
> - LC25000 (LC-Lung and LC-Colon): https://github.com/tampapath/lung_colon_image_set?tab=readme-ov-file
> - BRCAS: https://www.bracs.icar.cnr.it/
> - Camelyon17: https://camelyon17.grand-challenge.org/Data/
> - Camelyon16: https://camelyon16.grand-challenge.org/Data/
> - PathMMU: https://pathmmu-benchmark.github.io/#/

---

> > ### Author Response · Authors · 2024-11-24
> > **Response to Reviewer Y3n6 (2/2)**
> >
> > > ### **Only evaluated in CLIP model. Can be work in more pretrained Vision-Language model.**
> >
> > Thank you for your suggestion. We have added a ConvNeXt-base [1] convolution-based CLIP model trained by LAION [2] and a CLIPA-architecture model [3] . As shown in the following table, incorporating the PathGen-1.6M dataset significantly improves the performance of both new models, with improvements of 6.8% and 7.1%, respectively. This demonstrates the quality and effectiveness of the PathGen-1.6M dataset.
> >
> > | Model                                                | LC-Lung | LC-Colon | CRC100K | SkinCancer | Pcam | BACH | Osteo | WSSSLUAD | SICAPv2 | Average         |
> > | ---------------------------------------------------- | ------- | -------- | ------- | ---------- | ---- | ---- | ----- | -------- | ------- | --------------- |
> > | PathGen-CLIPA-L-336 _init（w/o PathGen-1.6M)         | 79.3    | 98.4     | 61.5    | 50.5       | 86.1 | 56.5 | 59.8  | 78.5     | 48.0    | 68.7            |
> > | PathGen-CLIPA-L-336 （w/ PathGen-1.6M)               | 94.3    | 99.4     | 68.4    | 64.9       | 88.8 | 62.0 | 63.7  | 80.9     | 57.0    | **75.5 (+6.8)** |
> > | PathGen-Convnext-base-w_320_init （w/o PathGen-1.6M) | 82.8    | 94.6     | 59.9    | 48.0       | 74.9 | 53.0 | 62.2  | 84.1     | 56.1    | 68.4            |
> > | PathGen-Convnext-base-w_320 （w/ PathGen-1.6M)       | 89.1    | 95.9     | 67.9    | 60.2       | 80.7 | 66.2 | 77.9  | 87.2     | 54.3    | **75.5 (+7.1)** |
> > | PathGen-CLIP-L_init                                  | 89.1    | 96.9     | 60.3    | 55.7       | 83.2 | 65.3 | 71.9  | 85.1     | 48.3    | 72.9            |
> > | PathGen-CLIP-L                                       | 89.8    | 99.3     | 78.0    | 70.6       | 88.2 | 71.5 | 74.6  | 82.2     | 63.5    | **79.7 (+6.8)** |
> >
> > [1] Liu Z, Mao H, Wu C Y, et al. A convnet for the 2020s[C]//Proceedings of the IEEE/CVF conference on computer vision and pattern recognition. 2022: 11976-11986.
> >
> > [2] Schuhmann C, Beaumont R, Vencu R, et al. LAION-5B: An open large-scale dataset for training next generation image-text models[C]//Thirty-sixth Conference on Neural Information Processing Systems Datasets and Benchmarks Track. 2022.
> >
> > [3] Li X, Wang Z, Xie C. An inverse scaling law for clip training[J]. Advances in Neural Information Processing Systems, 2024, 36.

---

> ### Author Response · Authors · 2024-11-26
>
> Dear Reviewer **```Y3n6```**,
>
>
>
> Thank you for your time and thoughtful consideration. We sincerely hope that we have addressed your concerns adequately. If there are any remaining issues or if you require further clarification, please do not hesitate to let us know.
>
> Once again, we truly appreciate your valuable feedback and support.

---

> > ### Comment · Reviewer_Y3n6 · 2024-11-29
> > **response to authors**
> >
> > Thank you very much for your response. Most of my concerns have been addressed, and this is a paper worthy of acceptance.

---

> > > ### Author Response · Authors · 2024-11-29
> > > **Thank you**
> > >
> > > Thank you for your time and valuable suggestions!

---

### Official Review · Reviewer_TdEJ · 2024-11-03

**Soundness:** 3
**Presentation:** 3
**Contribution:** 3
**Rating:** 8
**Confidence:** 4

**Summary:**

The paper proposes a new 1.6 million dataset of paired image-caption pathology data.
The authors propose a scalable way to construct the dataset using existing LMMs and refinement strategies.
They show this dataset is useful in developing a pathology specific CLIP model which performs better than existing domain specific and general purpose CLIP style models in zero-shot an few-shot problems
Finally they show that scaling dataset size and model size can lead to improvements in some tasks.

**Strengths:**

Originality:
The paper proposes a new way to generate quality image-caption pairs for pathology data.
It uses existing publicly available pathology image-caption data and the ability of newer general purpose LMMs like GPT-4 to generate detailed descriptions to create an initial dataset for training a LLaVA style captioning model.
It then uses this captioning model in conjunction with a revision and summary agent to generate synthetic image-caption data.
This approach of using existing data and LMMs to build a model and refine its outputs using other agents is interesting and not explored in the context of pathology.
Given the large nature of WSI images and significant redundancy and similarity of image content, the authors propose ways to construct a dataset of diverse image patches

Evaluation:
The authors evaluate the model on various zero-shot and few-shot patch-level tasks and on WSI-level prediction tasks and compare its performance against various existing models. The results with GigaPath a vision encoder are promising and show we can use existing vision encoders and improve their language capabilities using the captioning data.
They also qualitatively evaluate a small subset of the generated captions using expert pathologists.

Experiments:
The authors show various ablations around dataset construction which are valuable in understanding some of the limitations of using captioning models. The section scaling dataset and model size is interesting and shows some evidence around usefulness of scaling datasets and models using this approach.

All the data and models are publicly available which is great for the pathology community and this will also be the largest publicly available image caption dataset for pathology.

**Weaknesses:**

Clarity:
While I appreciate the authors covering a lot of ablations and experiments and describing the prompts, many of the design choices aren't clearly explained well. I've added some in the questions below.

Evaluation:
While the authors do compare with many older pathology VLM models, its unclear why they couldn't get access to the more recent CONCH which is publicly available on HuggingFace.
For WSI tasks while its helpful they added Gigapath, they don't compare against better publicly available pathology vision encoders like
H-Optimus.

Improvements:
Given the setup, its unclear how much of the improvements are coming from the new PathGen data and the refinement through revision agent, given they are one of the main contributions of the paper. i.e There isnt a comparison of the performance of the original PathGen-CLIP-L_init encoder with the improved PathGen-CLIP-L. Another comparison which would be useful is understanding with/out data generated using the revision agent.
It would be useful to highlight these as it helps understand how well such a setup can scale in generating synthetic data for iterative refinement.
Quality and Scale of Initial Dataset:
Its also unclear how important the scale of the detailed caption dataset 30K used for training PathGenLlava and the quality of it. Does scaling the dataset size and having some refinement here help? Have the authors checked the quality of captions generated by GPT-4 here and can provide some insight.

**Questions:**

Dataset Construction:
Its unclear why and how the subset of 700K samples was choosen from existing datasets to create PathGen_init.
For training the description agent, the authors mention using 10K initial image-caption pairs to generate 30K dataset, so are 3 captions generated per image?

Revision Agent:
What model is used for training the revision agent? Its also unclear what the inputs for revision agent are at inference? Does it take the generated caption and produce possible edits?


In figure 10, its unclear the w/ PathGen_init two stage performance doesnt vary when scale of PathGen data is varied? When scale is 0 it means all data is PathGen_init is that correct?

---

> ### Author Response · Authors · 2024-11-24
> **Response to Reviewer TdEJ (1/2)**
>
> Thank you for your valuable and insightful feedback, which has greatly helped us refine and improve our work. We have carefully addressed each of your suggestions one by one in our response.
>
> > ### Its unclear why and how the subset of 700K samples was choosen from existing datasets to create PathGen_init.
>
> This is an excellent question. The performance of prompt-based retrieval and integration with LLMs heavily depends on having a well-performing CLIP model. However, existing models are constrained by the quality and quantity of their training data, resulting in suboptimal multimodal capabilities. To overcome this limitation, we carefully curated and collected a larger vision-language dataset of 700K samples to train a more robust CLIP initial model. As shown in the table below, the PathGen-CLIP-L_init model, trained on this dataset, already surpasses the previous SOTA models.
>
> | **Model**           | **LC-Lung** | **LC-Colon** | **CRC100K** | **SkinCancer** | **Pcam** | **BACH** | **Osteo** | **WSSSLUAD** | **SICAPv2** | **Average** |
> | ------------------- | ----------- | ------------ | ----------- | -------------- | -------- | -------- | --------- | ------------ | ----------- | ----------- |
> | Previous  SOTA      | 88.9        | 94.3         | 55.3        | 35.1           | 72.5     | 46.8     | 69.2      | 85.1         | **48.3**    | 66.2        |
> | PathGen-CLIP-L_init | **89.1**    | **96.9**     | **60.3**    | **55.7**       | **83.2** | **65.3** | **71.9**  | **85.1**     | **48.3**    | **72.9**    |
>
> In constructing PathGen_init, we retained almost all of the data from PathCap, particularly as the authors had already performed some level of data cleaning. For the Quilt-1M and OpenPath datasets, we conducted an additional cleaning process to address instances where captions did not align well with the images. For example, captions in social media posts often emphasized the aesthetic qualities of the images or solicited diagnostic opinions from the public. Furthermore, there was some redundancy among the images. After this cleaning process, we preserved a total of 700K high-quality data points.
>
> > ### For training the description agent, the authors mention using 10K initial image-caption pairs to generate 30K dataset, so are 3 captions generated per image?
>
> We apologize if our wording caused any misunderstanding. As we mentioned, “we sample 10,000 image-caption pairs from PathCap, OpenPath, and Quilt-1M, respectively,” this means that 10K samples were taken from each of PathCap, OpenPath, and Quilt-1M, adding up to a total of 30K. We will revise our wording to make this clearer.
>
> > ### Revision Agent: What model is used for training the revision agent? Its also unclear what the inputs for revision agent are at inference? Does it take the generated caption and produce possible edits?
>
> We apologize for not expressing this clearly. Revise agent is trained using the LLaVA v1.5 framework. During the inference phase, the input consists of the description generated by a previous LMM description agent, paired with the corresponding image, forming a multimodal input.  Revise agent processes this input to produce a series of edits.
>
> > ### In figure 10, its unclear the w/ PathGen_init two stage performance doesnt vary when scale of PathGen data is varied? When scale is 0 it means all data is PathGen_init is that correct?
>
> We apologize for not explaining this more clearly. The two-stage approach with PathGen_init serves as a reference point, representing the final result of our current method, which is why its performance does not vary. For the orange line (with PathGen_init one-stage), a scale of 0 indeed means that all the data is from PathGen_init.
>
> Thank you for pointing this out, and we will include a more detailed explanation in the appendix.
>
> > ### Its unclear why they couldn't get access to the more recent CONCH which is publicly available on HuggingFace.
>
> Access to CONCH requires approval, and since we obtained access relatively late, we were unable to include experiments involving CONCH. However, we have now included experiments to benchmark CONCH in **General Response** of the rebuttal. **PathGen-CLIP-L** demonstrates significantly better performance across various downstream tasks compared to the CONCH. Please refer to the general response for more details.
>
> Access to CONCH requires approval, and since we obtained access relatively late, we were unable to include experiments involving CONCH.

---

> ### Author Response · Authors · 2024-11-24
> **Response to Reviewer TdEJ (2/2)**
>
> > ### Don't compare against better publicly available pathology vision encoders like H-Optimus.
>
> Thank you for the suggestion.  Our main contribution is the PathGen-1.6M dataset per se, which is why we chose `datasets and benchmarks` as our **Primary Area**.  The PathGen-1.6M dataset can be effectively applied to more advanced models (such as GigaPath and H-Optimus) to further enhance performance. As highlighted in Section 4.6, we demonstrated how PathGen-1.6M could fine-tune vision-only models like GigaPath, achieving the following capabilities:
>
> 1. The transformation of vision-only models into vision-language models, enabling multimodal capabilities.
> 2. Further improving their vision performance.
>
> Additionally, our model was trained solely on publicly available datasets, specifically **7,500** WSIs from TCGA, using significantly fewer patches and smaller hyperparameters (OpenAI CLIP-L). In contrast, models like H-Optimus were trained on **over 500,000 privately collected WSIs**, utilizing 4 times larger model VIG-G. Such data scale and resources are **inaccessible to researchers** relying solely on public datasets.
>
> > ### There isnt a comparison of the performance of the original PathGen-CLIP-L_init & Comparison which would be useful is understanding with/out data generated using the revision agent.
>
> | **Model**                       | **LC-Lung** | **LC-Colon** | **CRC100K** | **SkinCancer** | **Pcam** | **BACH** | **Osteo** | **WSSSLUAD** | **SICAPv2** | **Average** |
> | ------------------------------- | ----------- | ------------ | ----------- | -------------- | -------- | -------- | --------- | ------------ | ----------- | ----------- |
> | PathGen-CLIP-L_init             | 89.1        | 96.9         | 60.3        | 55.7           | 83.2     | 65.3     | 71.9      | **85.1**     | 48.3        | 72.9        |
> | PathGen-CLIP-L wo/ Revise agent | **91.2**    | 98.9         | 76.2        | 67.2           | 87.3     | 68.8     | 70.8      | 81.5         | 54.9        | 77.4        |
> | PathGen-CLIP-L                  | 89.8        | **99.3**     | **78.0**    | **70.6**       | **88.2** | **71.5** | **74.6**  | 82.2         | **63.5**    | 79.7        |
>
> Thank you for the suggestion. We have included the results for **PathGen-CLIP-L_init**, and **PathGen-CLIP-L** significantly outperforms it, with an average improvement of 6.8%. Notably, performance improved by 14.9% on SkinCancer, 17.7% on CRC, and 15.2% on SICAPv2. This highlights that **PathGen-1.6M** can significantly enhance the image-text alignment capabilities of CLIP.
>
> Additionally, without the Revise Agent, the performance of **PathGen-CLIP-L** suffers a 2.3% degradation. This demonstrates that the Revise Agent effectively corrects potential errors or hallucinations from the description agent.
>
> > ### Its also unclear how important the scale of the detailed caption dataset 30K used for training PathGenLlava and the quality of it. Does scaling the dataset size and having some refinement here help?
>
> Your suggestion is absolutely correct. We could investigate scaling up the detailed caption dataset to explore whether training a better description LMM agent could further improve model performance. However, the challenge lies in the fact that each time we scale up the detailed caption dataset, we need to **retrain and regenerate almost all the models and data, which results in substantial time and cost overhead**. We apologize that the rebuttal period was too short for us to conduct this experiment and we plan to address this in future work.
>
> However, to demonstrate that improving the quality of the dataset is effective from another perspective, we used our trained **PathGen-LLaVA model to refine the captions of the 700K PathGen_init dataset.** We observed that this refinement can lead to a 1.4% improvement in model performance. This indicates that a higher-quality image-caption dataset can enhance overall model performance.
>
> | **Model**                                    | **LC-Lung** | **LC-Colon** | **CRC100K** | **SkinCancer** | **Pcam** | **BACH** | **Osteo** | **WSSSLUAD** | **SICAPv2** | **Average** |
> | -------------------------------------------- | ----------- | ------------ | ----------- | -------------- | -------- | -------- | --------- | ------------ | ----------- | ----------- |
> | PathGen-CLIP-L                               | 89.8        | 99.3         | **78.0**    | 70.6           | 88.2     | **71.5** | 74.6      | 82.2         | **63.5**    | 79.7        |
> | PathGen-CLIP-L (w/ refined **PathGen_init)** | **92.9**    | **99.7**     | 77.7        | **73.5**       | **89.9** | 70.3     | **78.2**  | **86.5**     | 61.3        | **81.1**    |

---

> ### Author Response · Authors · 2024-11-26
>
> Dear Reviewer **```TdEJ```**,
>
>
>
> Thank you once again for your time and thoughtful consideration. We hope that we have addressed your concerns, and would be grateful if you could share your feedback. We are more than happy to provide any further clarification or engage in additional discussion if needed.

---

> > ### Comment · Reviewer_TdEJ · 2024-11-26
> >
> > Thanks to the authors for addressing the various concerns and running additional experiments to generate more evidence.
> >
> > >... For the Quilt-1M and OpenPath datasets, we conducted an additional cleaning process to address instances where captions did not align well with the images ...
> >
> > I think its worth highlighting this in the paper and if you can share this cleaned dataset publicly, that would be helpful.
> >
> > Some of the additional ablations that you have shown here comparing the original PathGen-CLIP-L_init are useful and should be added to the paper.

---

> ### Author Response · Authors · 2024-11-26
>
> Dear Reviewer **```TdEJ```**,
>
> Thank you very much for taking the time to read our response, and we also appreciate your further suggestions.
>
> Regarding the mention of data cleaning, we have made updates in lines 149-152, which are highlighted in red in the manuscript. **These cleaned datasets will be made fully public**. However, since we do not directly own the rights to distribute these datasets, we will provide the image IDs, and researchers can extract the images from the original datasets.
>
> As for the additional experiments, such as the comparison to PathGen-CLIP-L$_{init}$, we have included these results in Appendix section B5.1, Table 9, along with some other ablations.
>
> If you feel that our responses resolve the concerns raised, we would be grateful if you could kindly consider this in your evaluation. Please feel free to let us know if you have any further questions or would like additional clarifications.
>
> Thank you once again for your time and reply!

---

> > ### Comment · Reviewer_TdEJ · 2024-11-26
> >
> > Thanks for clarifying, I have updated my score

---

> > > ### Author Response · Authors · 2024-11-26
> > >
> > > We sincerely appreciate your time and thoughtful consideration. Thank you once again!

---

### Official Review · Reviewer_wDqp · 2024-11-05

**Soundness:** 4
**Presentation:** 3
**Contribution:** 3
**Rating:** 8
**Confidence:** 5

**Summary:**

PathGen-1.6M represents a significant advancement in pathology AI, introducing the largest high-quality pathology image-text dataset created through multi-agent collaboration. The approach leverages whole slide images from TCGA to extract representative patches and generate accurate, detailed captions, achieving 88-90% accuracy validated by pathologists. The resulting models, PathGen-CLIP and PathGen-LLaVA, demonstrate superior performance across various tasks including zero-shot classification, few-shot learning, and whole slide image analysis, outperforming existing models including GPT-4V. This work provides a scalable pathway for generating high-quality pathology data and developing more capable AI models for clinical applications.

**Strengths:**

- Well written introduction
- Good literature survey on LMMs and Pathology--CLIPs
- Brilliant idea on how to develop the revise LLM agent
- Representative patch selection using GPT-4 is an excellent idea
- Interesting approach to select patches through clustering but modulating the number of clusters as the square root of the size of a slide
- Strong evaluation pipeline across a range of tasks

**Weaknesses:**

- It's an overkill to say that this is an agent based system. Agent based systems are autonomously operating on a set of predefined rules and behaviors, while this approach appears to be more of a sequential pipeline with different models performing specific tasks rather than truly autonomous agents interacting with each other.
- Lit survey on multi-agent architectures could be expanded
- Usage of GPT-4s internal knowledge about the morphology of an organ is a good idea but deters diversity of patches collected. Also unclear if those prompts are well represented in the CLIP training dataset. Additional details on how many prompts obtained for each WSI will help the reader. It's unclear if each patch is matched against 2 prompts (report and attribute based) or more?
- Added details on motivations for design choices such as including both prompt and image retrieval will help.
- The methods section needs more details and fleshing out the writing will help; I think this is also generally true for most of the paper
- While not the goal of the paper, it will help to include fully supervised baselines as well to educate the readers of the gap with CLIP like models
- Details on how the instruction tuning data was curated are not provided

**Questions:**

- How many real pathology report findings did you'll extract in section 3 and how did you verify the quality of it
- How do you evaluate the performance of the images retrieved for the prompts in section 3.1?
- How many prompts do you use for each WSI?
- How did you arrive at the design choice of using both prompt-based and clustering-based retrieval?
- Why chose a threshold of 0.88 for similarity?
- Is similarity computed globally in all the extracted patches across slides?
- Does the revision agent have a no-operation capability as well? What happens when a correct description is passed to the revision agent?
- What does first and second stage training in PathGen-CLIP mean?
- Why did you use different datasets for zero and few shot experiments? As an example, Camelyon is not included in zero-shot examples

---

> ### Author Response · Authors · 2024-11-24
> **Response to Reviewer wDqp (1/3)**
>
> Thank you for your thorough review and detailed suggestions. We appreciate your recognition of the strengths of our approach. Regarding your questions, we address them one by one below:
>
> > ### How many real pathology report findings did you'll extract in section 3 and how did you verify the quality of it
>
> Thank you for your feedback. To clarify, the pathology report findings for WSI were extracted as **1-3 blocks of paragraphs containing the findings, rather than segmented into independent entries**. This is because the entire report often exceeds CLIP's 77-token limit, so the portions of the report that exceeded the length were divided into two to three sub-paragraphs to ensure all relevant information was retained without truncation.
>
> To ensure quality, we conducted multiple rounds of prompt engineering. **In each iteration, a human pathologist reviewed at least ten reports cleaned by GPT-4o. This iterative process involved 3-4 rounds of refinement and expert feedback,** culminating in a robust prompt that enabled GPT-4o to generate cleaned reports closely aligning with pathologists' expectations.
>
> > ### How do you evaluate the performance of the images retrieved for the prompts in section 3.1?
>
> We involved pathologists to manually evaluate the retrieved image patches to ensure they met expectations. Specifically,
>
> - Pathologists randomly reviewed 3 WSIs (as reviewing a large number of patches is highly time-consuming, it was not feasible to evaluate all WSIs).
> - They first identified observable findings in the predefined prompts and evaluated whether the retrieved patches captured these findings.
> - The evaluation revealed that **71.1%** of the patches matched the findings, when representative patches from clustering-based retrieval were included, this percentage increased to **80.3%**.
>
> > ### How many prompts do you use for each WSI?
>
> For each WSI, we generate 1-3 findings-based prompts and 20 attribute-based prompts, resulting in a total of 21-23 prompts per WSI.
>
> > ### How did you arrive at the design choice of using both prompt-based and clustering-based retrieval?
>
> This is a very good question. We combined prompt-based and clustering-based retrieval to **balance diversity and representativeness in the retrieved patches.** Prompt-based retrieval utilizes high-quality WSI reports to **focus on diagnostically critical patches**, while clustering-based retrieval **ensures semantic diversity** by capturing a variety of features across the WSI. **This combination also addresses individual limitations,** such as the prompt-based method's occasional tendency to focus entirely on irrelevant artifacts like doctor-annotated black markings on WSIs. Clustering-based retrieval can avoid this by retrieving other relevant patches.
>
> > ### Why chose a threshold of 0.88 for similarity?
>
> This is an empirical result achieved by inviting pathologists to evaluate different thresholds. We adjusted the threshold to ensure the filtered images maintained appropriate distinctiveness.
>
> > ### Is similarity computed globally in all the extracted patches across slides?
>
> Similarity is calculated **within a single WSI**. This approach is chosen because patches within the same WSI are more similar compared to patches across different WSIs. Computing similarity globally across slides could risk over-deleting certain samples.
>
> > ### Does the revision agent have a no-operation capability as well? What happens when a correct description is passed to the revision agent?
>
> Yes, absolutely. During training, we retained approximately one-third of the data without any modifications. The revision agent outputs a list of operations, and when a correct description is passed to the revision agent, its output is simply an empty list.
>
> > ### What does first and second stage training in PathGen-CLIP mean?
>
> The first stage involves pretraining PathGen-CLIP using only PathGen-1.6M, followed by fine-tuning in the second stage with PathGen_init, which yields the best performance. For a detailed explanation of this approach, please kindly refer to Appendix Section B.3.

---

> ### Author Response · Authors · 2024-11-24
> **Response to Reviewer wDqp (2/3)**
>
> > ### Why did you use different datasets for zero and few shot experiments? As an example, Camelyon is not included in zero-shot examples
>
> Thank you for pointing this out. Since datasets like BACH and SkinCancer cannot provide the full 256 shots per class due to their size, and given space constraints in the article, we only included results for four datasets.
>
> Here, we present the complete few-shot results for the remaining five datasets, using the largest feasible number of shots (2, 4, 8, 16, 32, 64, 128, and 256) based on the dataset size. As shown in the following tables, the PathGen-CLIP series demonstrates significantly better performance compared to previous CLIP-based pathology models.
>
> | **Dataset: BACH**     | **2**    | **4**    | **8**    | **16**   | **32**   | **64**   | **128** | **256** |
> | --------------------- | -------- | -------- | -------- | -------- | -------- | -------- | ------- | ------- |
> | PLIP                  | 49.4     | 57.8     | 64.0     | 67.4     | 72.3     | 78.2     | -       | -       |
> | PathCLIP              | 56.7     | 64.0     | 78.2     | 72.3     | 76.9     | 78.2     | -       | -       |
> | QuiltNet              | 50.0     | 57.1     | 61.8     | 66.5     | 71.1     | 76.6     | -       | -       |
> | PathGen-CLIP (ours)   | 59.5     | 71.7     | 77.5     | 83.4     | 89.8     | 92.3     | -       | -       |
> | PathGen-CLIP-L (ours) | **70.1** | **75.0** | **81.3** | **86.0** | **90.4** | **93.3** | -       | -       |
>
> | **Dataset: SICAP-v2** | **2**    | **4**    | **8**    | **16**   | **32**   | **64**   | **128**  | **256**  |
> | --------------------- | -------- | -------- | -------- | -------- | -------- | -------- | -------- | -------- |
> | PLIP                  | 49.1     | 57.8     | 56.4     | 63.9     | 67.1     | 67.3     | 69.7     | 70.5     |
> | PathCLIP              | 52.1     | 55.7     | 60.9     | 63.7     | 67.4     | 70.3     | 72.2     | 74.0     |
> | QuiltNet              | 55.1     | 60.4     | 59.8     | 66.0     | 70.7     | 71.9     | 72.9     | 74.0     |
> | PathGen-CLIP (ours)   | 57.3     | **65.6** | **70.1** | **74.2** | **76.3** | 75.6     | 76.4     | 76.2     |
> | PathGen-CLIP-L (ours) | **61.3** | 65.4     | 69.6     | 73.3     | 76.1     | **77.1** | **77.6** | **78.9** |
>
> | **Dataset: Osteo**    | **2**    | **4**    | **8**    | **16**   | **32**   | **64** | **128**  | **256**  |
> | --------------------- | -------- | -------- | -------- | -------- | -------- | ------ | -------- | -------- |
> | PLIP                  | 74.0     | 79.0     | 84.7     | 86.2     | 88.2     | 89.5   | 91.0     | 92.4     |
> | PathCLIP              | 76.7     | 81.4     | 82.8     | 86.1     | 87.0     | 89.2   | 91.1     | 93.0     |
> | QuiltNet              | 66.2     | 72.8     | 77.4     | 82.2     | 87.2     | 89.6   | 91.4     | 92.6     |
> | PathGen-CLIP (ours)   | 74.6     | 85.1     | **87.6** | **90.5** | 92.8     | 94.5   | **95.9** | 96.6     |
> | PathGen-CLIP-L (ours) | **80.0** | **85.4** | 86.8     | 89.5     | **93.1** | 94.7   | 95.8     | **96.7** |
>
> | **Dataset: SkinCancer** | **2**    | **4**    | **8**    | **16**   | **32**   | **64**   | **128** | **256** |
> | ----------------------- | -------- | -------- | -------- | -------- | -------- | -------- | ------- | ------- |
> | PLIP                    | 63.0     | 70.9     | 75.7     | 80.8     | 83.1     | 85.4     | -       | -       |
> | PathCLIP                | 62.1     | 67.0     | 73.7     | 77.6     | 81.4     | 84.2     | -       | -       |
> | QuiltNet                | 63.7     | 69.4     | 77.0     | 80.4     | 84.1     | 85.9     | -       | -       |
> | PathGen-CLIP (ours)     | **71.0** | 77.8     | 83.2     | 86.6     | 88.8     | 90.0     | -       | -       |
> | PathGen-CLIP-L (ours)   | **71.0** | **78.7** | **86.4** | **89.3** | **90.9** | **91.8** | -       | -       |
>
> | **Dataset: PatchCamelyon** | **2**    | **4**    | **8**    | **16**   | **32**   | **64**   | **128**  | **256**  |
> | -------------------------- | -------- | -------- | -------- | -------- | -------- | -------- | -------- | -------- |
> | PLIP                       | 76.0     | 76.2     | 79.6     | 87.7     | 91.2     | 92.0     | 92.9     | 93.4     |
> | PathCLIP                   | 78.8     | 79.9     | 82.1     | 89.3     | 91.1     | 93.9     | 95.0     | 95.1     |
> | QuiltNet                   | 79.1     | 80.2     | 83.5     | 90.2     | 91.0     | 94.6     | 94.9     | 95.4     |
> | PathGen-CLIP (ours)        | 88.9     | 84.6     | 87.9     | 94.2     | 94.5     | 94.8     | 95.6     | 96.0     |
> | PathGen-CLIP-L (ours)      | **89.3** | **89.2** | **93.4** | **93.4** | **95.0** | **95.6** | **96.3** | **96.5** |

---

> ### Author Response · Authors · 2024-11-24
> **Response to Reviewer wDqp (3/3)**
>
> > ### Details on how the instruction tuning data was curated are not provided
>
> Thanks for pointing is out. We apologize for not clearly explaining the process of generating the instruction-tuning data. Specifically, we randomly sampled 95K and 105K non-overlapping examples and used the prompts (shown in Figure 15 and Figure 16 in the appendix) to prompt GPT-4 to generate instruction-tuning data based on the sampled image-caption pairs. We will  clarify this in the main paper in the future revision.
>
> > ### While not the goal of the paper, it will help to include fully supervised baselines as well to educate the readers of the gap with CLIP like models
>
> Thank you for the suggestion. We provided fully supervised baselines as follows.
>
> - As shown, there remains a significant gap between zero-shot performance and full fine-tuning. However, the advancements of PathGen-CLIP-L bring the performance of pathology-specific CLIP closer to that of full fine-tuning.
> - Moreover, the full fine-tuning performance of PathGen-CLIP-L shows a substantial improvement over OpenAI-CLIP-L and is comparable to GigaPath, despite the latter benefiting from significantly larger training datasets and models.
>
> |                                | **WSSSLUAD** | **Lung** | **Colon** | **PatchCamelyon** |
> | ------------------------------ | ------------ | -------- | --------- | ----------------- |
> | Previous SOTA (Zero-shot)      | 85.1         | 88.9     | 94.3      | 72.5              |
> | PathGen-CLIP-L (Zero-shot)     | 82.2         | 89.8     | 99.3      | 88.2              |
> | OpenAI-CLIP-L (Full Finetune)  | 95.9         | 99.7     | 100.0     | 93.5              |
> | GigaPath-G (Full Fnetune)      | 97.1         | 100.0    | 100.0     | 96.9              |
> | PathGen-CLIP-L (Full Finetune) | 97.7         | 100.0    | 100.0     | 97.0              |
>
> > ### Referring to this caption generation system as agent-based might be somewhat exaggerated.
>
> Thank you for your valuable feedback. While we acknowledge that our system currently functions more like a sequential pipeline, it can still be conceptualized as a simple agent-based framework, where each module operates autonomously within its defined scope (e.g., the Revise Agent can independently decide whether to modify captions based on the received image-text input). Similar modular designs have been employed in works like MORA [1]. We believe that expanding components, such as the Revise Agent or LMM Agent, into more interactive roles—such as "round-table discussions"—would further enhance the system’s agent-based nature, and we leave this for future work.
>
> [1] Yuan Z, Liu Y, Cao Y, et al. Mora: Enabling generalist video generation via a multi-agent framework[J]. arXiv preprint arXiv:2403.13248, 2024.

---

> ### Author Response · Authors · 2024-11-26
>
> Dear Reviewer **```wDqp```**,
>
> We hope that we have addressed your concerns, and would be grateful if you could share your feedback. We are more than happy to provide any further clarification or engage in additional discussion if needed.
>
> Many thanks for your time and effort!

---

> > ### Comment · Reviewer_wDqp · 2024-11-26
> >
> > Thank you for your responses - they are quite helpful. I will increase my score.

---

> > > ### Author Response · Authors · 2024-11-26
> > >
> > > We greatly appreciate your recognition. Thank you once again for your time and valuable suggestions!

---

### Author Response · Authors · 2024-11-24
**General Response: Sincere Thanks and Comparison with CONCH Results**

We sincerely thank all the reviewers for your thoughtful and detailed feedback, as well as the time and effort you have devoted to reviewing our manuscript. We have addressed each reviewer’s comments and suggestions, their valuable insights have significantly improved the quality and clarity of our work.

**We have uploaded the revised manuscript, and to make the revisions clear, we have highlighted the updated sections in red.**

In the general response, as two reviewers mentioned the need for comparisons with CONCH, we have supplemented the experiments with zero-shot classification, few-shot classification, and whole-slide image classification tasks compared to CONCH.

As shown in the following tables,

- **PathGen-CLIP-L** significantly outperforms CONCH in zero-shot evaluation, achieving an average of 79.7% compared to CONCH's 68.7%.
- In few-shot evaluation, **PathGen-CLIP-L** surpasses CONCH in the most of shots.
- Moreover, for whole-slide image tasks, **PathGen-CLIP-L** exceeds CONCH across all metrics except the AUC on the BRACS dataset, further demonstrating the effectiveness of **PathGen-CLIP-L**.

### **1) Zero-shot evaluation:**

| **Model**      | **LC-Lung** | **LC-Colon** | **CRC100K** | **SkinCancer** | **Pcam** | **BACH** | **Osteo** | **WSSSLUAD** | **SICAPv2** | **Average** |
| -------------- | ----------- | ------------ | ----------- | -------------- | -------- | -------- | --------- | ------------ | ----------- | ----------- |
| CONCH          | 74.7        | 97.9         | 59.4        | 63.2           | 78.7     | 58.3     | 73.5      | 79.8         | 33.0        | 68.7        |
| PathGen-CLIP-L | **89.8**    | **99.3**     | **78.0**    | **70.6**       | **88.2** | **71.5** | **74.6**  | **82.2**     | **63.5**    | **79.7**    |

### **2) Few-shot evaluation:**

| Dataset: LC-Colon | **8**     | **16**    | **32**   | **64**    | **128**   | **256**   |
| --------------------- | --------- | --------- | -------- | --------- | --------- | --------- |
| CONCH                 | 99.7      | 99.8      | 99.8     | 99.9      | **100.0** | **100.0** |
| PathGen-CLIP-L        | **100.0** | **100.0** | **99.9** | **100.0** | **100.0** | **100.0** |

| Dataset: LC-Lung | **8**    | **16**   | **32**   | **64**   | **128**  | **256**  |
| -------------------- | -------- | -------- | -------- | -------- | -------- | -------- |
| CONCH                | 96.4     | **97.7** | **98.0** | **98.4** | 98.6     | 99.0     |
| PathGen-CLIP-L       | **96.5** | 97.5     | 97.8     | 98.1     | **98.7** | **99.1** |

| Dataset: WSSSLUAD | **8**    | **16**   | **32**   | **64**   | **128**  | **256**  |
| --------------------- | -------- | -------- | -------- | -------- | -------- | -------- |
| CONCH                 | 93.2     | 93.5     | 94.5     | 94.8     | 95.2     | 95.5     |
| PathGen-CLIP-L        | **93.9** | **95.0** | **95.2** | **95.3** | **95.7** | **95.7** |

|  | **8**    | **16**   | **32**   | **64**   | **128**  | **256**  |
| -------------------------- | -------- | -------- | -------- | -------- | -------- | -------- |
| CONCH                      | **94.6** | 94.0     | 95.7     | 96.3     | 96.4     | **96.6** |
| PathGen-CLIP-L             | 94.5     | **94.4** | **95.9** | **96.4** | **96.6** | **96.6** |

### **3) WSI-classification evaluation:**

|            |                | **ABMIL** |          | **ACMIL** |          |
| ---------- | -------------- | --------- | -------- | --------- | -------- |
|            |                | AUC       | F1       | AUC       | F1       |
| CAMELYON16 | PathGen-CLIP-L | **95.8**  | **94.3** | **97.4**  | **94.5** |
|            | CONCH          | 95.2      | 93.9     | 97.2      | 94.4     |
| CAMELYON17 | PathGen-CLIP-L | **87.9**  | **58.6** | **92.0**  | **58.4** |
|            | CONCH          | 86.4      | 55.0     | 87.5      | 56.3     |
| BRACS      | PathGen-CLIP-L | 87.2      | **66.6** | 88.4      | **66.9** |
|            | CONCH          | **90.0**  | 62.0     | **88.7**  | 66.1     |
| Average    | PathGen-CLIP-L | 90.3      | **73.2** | **92.6**  | **73.3** |
|            | CONCH          | **90.5**  | 70.3     | 91.1      | 72.7     |

For other concerns, please refer to our detailed responses to your individual comments for more information.

---

### Meta-Review · Area_Chair_KzvL · 2024-12-20

**Metareview:**

Strengths
1. **Novelty and Impact**: The manuscript presents a significant contribution by introducing the PathGen-1.6M dataset and PathGen-CLIP model, which demonstrate strong performance in pathology-related tasks.
2. **Methodology**: The use of a multi-agent pipeline for generating and refining image-caption pairs is innovative and aligns well with the paper's objectives.
3. **Evaluation**: Comprehensive experiments with ablations and evaluations, alongside public release of data and models, enhance the study's impact and utility for the pathology community.

Key Issues to Address

- Several design choices in the methods are not well explained. Clearer motivations and rationale for including prompts, image retrieval, and other steps are necessary.
- Details on the instruction tuning data and the number of prompts per WSI are incomplete.
- Missing baselines, especially with models like CONCH, UNI, and H-Optimus, reduce the comprehensiveness of comparisons.
- The study lacks evaluation of fully supervised baselines and critical comparisons, such as between PathGen-CLIP-L_init and PathGen-CLIP-L or the utility of the revision agent.
- The importance of the initial 30K dataset for PathGenLlava and the quality of GPT-4-generated captions should be further analyzed.
- The manuscript positions the pipeline as agent-based, but it operates more as a sequential model framework. Expanding on multi-agent system principles or reframing terminology could resolve this.

**Additional Comments On Reviewer Discussion:**

The authors have clearly clarified the issues raised by the reviewers. The explanations in the rebuttal look reasonable and correct to me.

---

### Decision · Program_Chairs · 2025-01-22

Accept (Oral)